# Fast response of cold ice-rich permafrost in northeast Siberia to a warming climate

Jan Nitzbon [1,2,3✉], Sebastian Westermann[3], Moritz Langer[1,2], Léo C. P. Martin [3], Jens Strauss [1], Sebastian Laboor [1] & Julia Boike [1,2]

The ice- and organic-rich permafrost of the northeast Siberian Arctic lowlands (NESAL) has been projected to remain stable beyond 2100, even under pessimistic climate warming scenarios. However, the numerical models used for these projections lack processes which induce widespread landscape change termed thermokarst, precluding realistic simulation of permafrost thaw in such ice-rich terrain. Here, we consider thermokarst-inducing processes in a numerical model and show that substantial permafrost degradation, involving widespread landscape collapse, is projected for the NESAL under strong warming (RCP8.5), while thawing is moderated by stabilizing feedbacks under moderate warming (RCP4.5). We estimate that by 2100 thaw-affected carbon could be up to three-fold (twelve-fold) under RCP4.5 (RCP8.5), of what is projected if thermokarst-inducing processes are ignored. Our study provides progress towards robust assessments of the global permafrost carbon–climate feedback by Earth system models, and underlines the importance of mitigating climate change to limit its impacts on permafrost ecosystems.

[1] Permafrost Research Section, Alfred Wegener Institute Helmholtz Centre for Polar and Marine Research, Telegrafenberg A45, 14473 Potsdam, Germany. [2] Geography Department, Humboldt-Universität zu Berlin, Unter den Linden 6, 10099 Berlin, Germany. [3] Department of Geosciences, University of Oslo, Sem Sælands vei 1, 0316 Oslo, Norway. ✉email: jan.nitzbon@awi.de

Today's permafrost landscapes have been shaped by climatic, geomorphic and ecological processes during the Last Glacial Period and the Holocene[1–5]. In unglaciated regions, climate-driven accumulation and melting of ground ice have been key processes in the evolution of these landscapes, resulting in history-dependent landscapes and landforms with distinct ground ice distributions[6–9]. There is emerging evidence that the present-day ground ice distribution governs permafrost thaw pathways, and thus how landscapes evolve in the future[10–13] —particularly given the expected warming of the Arctic climate[14,15]. Thawing of ice-rich permafrost and melting of massive ground ice induce landscape change termed thermokarst, which results in characteristic landforms across ice-rich permafrost terrain[16] (see Supplementary Notes 1 for definitions). In the continuous permafrost zone, thermokarst is expressed in the transition from low-centred to high-centred ice-wedge polygons[12,17], or the formation of thaw lakes and thermo-erosional gullies[16,18,19], thereby causing landscape-scale feedbacks on hydrology and carbon decomposition[12,20,21]. In contrast to the gradual thawing of permafrost in ice-poor terrain, thermokarst processes can cause severe permafrost degradation within few years or decades, and have thus been referred to as rapid or abrupt thaw[11,22,23]. The contribution of thermokarst processes to global-scale permafrost degradation in the future is highly uncertain[11,22]. Recent efforts using simple conceptual models allow first-order estimates and emphasise the global relevance of abrupt thaw in thermokarst terrain[23], but are at the same time limited by strong assumptions on model parameters that mask the underlying physical processes. To date, process-based Earth system models (ESMs) lack the structure and physical processes relevant to represent thermokarst, and might thus substantially underestimate future permafrost degradation and the permafrost carbon–climate feedback.

Permafrost in the northeast Siberian Arctic lowlands (NESAL, Fig. 1a, see "Methods" for definition) is highly susceptible to thermokarst[18], as the landscapes' history led to abundance of ice- and organic-rich permafrost deposits[3,24] (Fig. 1b; Supplementary Fig. 1). The NESAL comprise large parts of the Yedoma domain[25,26], which has been hypothesised to exhibit large-scale tipping behaviour under strong regional climate warming[27]. Permafrost deposits in the NESAL are estimated to store about 100 GtC, corresponding to an increase in atmospheric $CO_2$ of about 24 ppm if it were all to be released into the atmosphere (Supplementary Methods 1). Despite recent warming trends observed in boreholes (about 0.9 °C per decade[28,29]), permafrost temperatures in the NESAL are yet amongst the coldest in the Arctic with observed and simulated present-day mean annual ground temperatures lying mostly within the range −8 to −12 °C[30]. In projections of ESMs, the NESAL are one of the most stable permafrost regions, with near-surface permafrost largely remaining thermally stable beyond 2100, even under the strong RCP8.5 warming scenario[31–34]. However, these models do not take into account thermokarst-inducing processes and associated feedbacks, which can be expected to occur in the NESAL due to the abundance of massive ground ice in form of ice wedges[11,12,22,23], making the model projections highly questionable.

Recent developments constitute significant progress towards the process-based simulation of subgrid-scale heterogeneity and ground subsidence[10,13,35–37], allowing a more realistic evaluation of ice-rich permafrost thaw dynamics. Here, we extend and apply the CryoGrid 3 permafrost model[13] to investigate how the present-day ground ice distribution in the NESAL, which is inherited from the landscape history, affects pathways of landscape evolution, the magnitude and pace of permafrost degradation, and the amount of currently freeze-locked permafrost carbon that becomes subject to thawed conditions, in the course of the twenty-first century. Our focus here is on vast lowlands (covering about 493,000 km$^2$) that are underlain by massive ice wedges in the subsurface, as wedge ice constitutes the dominant

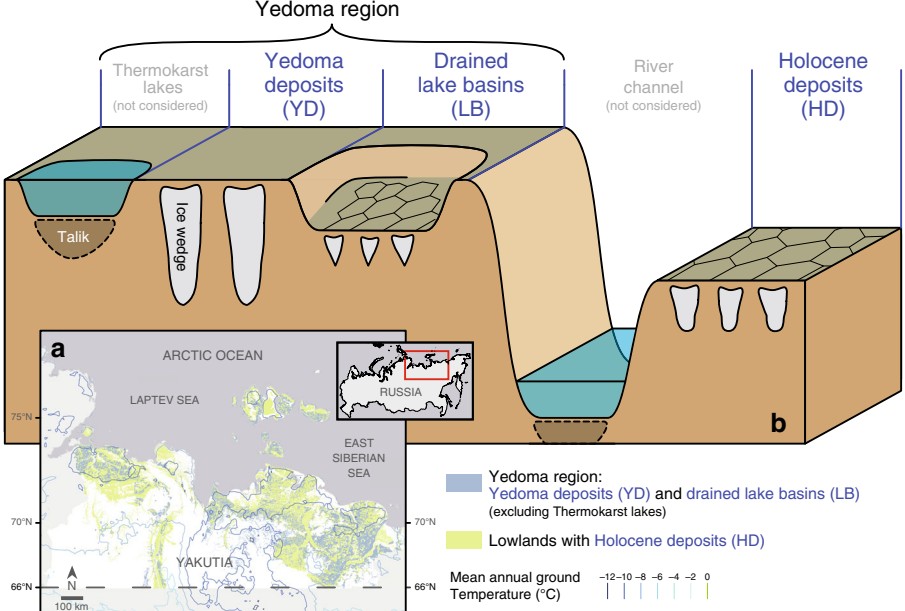

**Fig. 1 Map and schematic of ice-rich permafrost landscapes. a** Map showing the extent of ice-rich lowlands and mean annual ground temperatures[30] in northeast Siberia. See Supplementary Fig. 2 for an enlarged map of the study region. **b** Climate-driven landscape evolution during the last glacial period and the Holocene has led to diverse types of ice-rich permafrost landscapes. At present, ice wedges are the most common form of excess ground ice, but there is large variability in the volume and thickness of wedge ice. We assumed ice wedges to increase in lateral size and depth from drained lake basins (LB), over Holocene deposits (HD), to undisturbed Yedoma deposits (YD). While LB and HD feature active ice-wedge polygons with a patterned surface microtopography, YD host relict ice-wedge polygons which are covered by younger deposits. Note that we excluded from our analysis old thermokarst lakes and large water bodies already existing in the study region. See Supplementary Fig. 1 for an extended version of the schematic.

type of excess ground ice in the NESAL[3,38] (see Supplementary Notes 2). Consequently, the initiation and development of thermokarst can be assumed to involve melting of ice wedges at the microscale. Acknowledging the landscape evolution throughout the Holocene, we distinguish three major types of present-day landscapes, with marked differences in the wedge-ice volumes and the thickness of ice-rich deposits (Fig. 1b). Intact Late Pleistocene deposits within the Yedoma region are underlain my wide and deep ice wedges (Yedoma deposits, YD), while large parts of the Yedoma region have been affected by degradation processes during the Holocene, involving the formation of thermokarst lakes and their drainage. Refrozen drained lake basins (LB) are underlain by thin and shallow ice wedges. Active ice-wedge polygons are also abundant in lowlands outside the Yedoma region, which we designated as Holocene deposits (HD). To broadly explore possible landscape responses, the present-day landscapes are subjected to different hydrological conditions (water-logged versus well-drained) and future climate scenarios following different representative concentration pathways (RCP2.6, RCP4.5 and RCP8.5). Our model explicitly takes into account the spatial heterogeneity in surface topography and subsurface stratigraphies of ice-rich permafrost terrain, ground subsidence due to melting of excess ice, as well as feedbacks exerted through small-scale lateral fluxes of heat, water and snow. In addition, lateral sediment transport is reflected in our model, which is a key processes for the stabilisation of ice wedges after initial degradation[39]. To allow for comparison with the simplistic representation of permafrost in ESMs, we conducted reference runs which do not take into account thermokarst-inducing processes, such as excess ice melt and small-scale lateral fluxes. We demonstrate that our numerical modelling approach can retrace a multitude of landscape evolution and degradation pathways characteristic to ice-wedge terrain in the continuous permafrost zone. Projected permafrost degradation in the NESAL during the twenty-first century is substantially increased when thermokarst-inducing processes are taken into account. The response of ice-rich terrain differs considerably for different future warming scenarios, ranging from mostly stable landscapes (RCP2.6) to widespread landscape collapse (RCP8.5). We find that lateral sediment transport moderates thaw after initial ice-wedge degradation, leading to stabilised landscapes by 2100 under RCP4.5. For RCP8.5, we find that substantial amounts of the NESAL's organic carbon stocks might be affected by thaw within the present century, despite the projected stability of the region in previous ESM projections. This study emphasises the necessity of representing thermokarst-inducing processes in ESMs and provides significant progress towards achieving this goal. More broadly, our results underline the importance of mitigating climate change if we are to limit its impacts on permafrost ecosystems.

## Results and discussion

**Landscape evolution.** Our numerical model uses the concept of laterally coupled tiles[35–37] to represent the spatial heterogeneity in surface and subsurface characteristics of ice-wedge terrain, which is typically characterised by polygonal patterned ground (see "Methods"; Supplementary Methods 2). We classified the geomorphological state of the landscape according to the relative positions of the tiles' soil surface altitudes, thereby distinguishing between relict polygons (RP), low-centred polygons (LCP), intermediate-centred polygons (ICP), high-centred polygons (HCP) and water bodies (WB) (see "Methods" for definitions). For each type of ice-wedge terrain (lake basins (LB), Holocene deposits (HD) and Yedoma deposits (YD)), we assessed the future landscape evolution, starting from the representative present-day

state of the landscape (Fig. 2b, c; Supplementary Fig. 3; Supplementary Movies). Note that while our model allows for the formation of surface water bodies through ice-wedge thermokarst, the evolution of already existing old thaw lakes is not considered in this study.

Under the RCP2.6 scenario, all landscape types (LB, HD, YD) remained stable throughout the simulation period, with the exception of water-logged YD where shallow surface water bodies formed. We thus restrict the following analysis of the landscape evolution to the warming scenarios RCP4.5 and RCP8.5. The simulations for LB and HD were initialised with undegraded LCPs, featuring water-covered centres and elevated rims overlaying intact ice wedges (Fig. 2b; Supplementary Fig. 4a). Within the simulation period, ice-wedge degradation occurred and altered the initial landscape configuration under both warming scenarios (RCP4.5 and RCP8.5), and irrespective of the hydrological conditions (Fig. 2a). However, the initiation of ice-wedge degradation, which is indicated by the transition from LCP to ICP microtopography, was found to occur about two decades earlier under RCP8.5 than under RCP4.5, as well as about two decades earlier under water-logged compared with well-drained settings. Hence, the LB and HD landscapes were most stable under RCP4.5 and well-drained conditions, for which the simulated landscape by 2100 showed little sign of permafrost degradation (Fig. 2d). Under RCP8.5 and well-drained conditions, in turn, more substantial melting of ice wedges was simulated, indicated by subsiding rims of polygons and the resulting development of HCPs (Fig. 2e). When drainage was impeded, strong warming (RCP8.5) led to a collapse of ice-wedge polygons and the formation of surface water bodies (Fig. 2f). These water bodies are the initial stage for the formation of larger thermokarst lakes. In the RCP8.5 simulations, water bodies reach mean depths of ~1 m (LB), 2 m (HD) and 4 m (YD) within a few decades, and cause the development of continuously unfrozen zones (taliks) that were 3 m (LB) to 5 m (YD) thick (Supplementary Fig. 5m, n, o). Notably, the high-centred topography is preserved at the bottom of the water bodies, indicating that the lateral sediment transport does not keep pace with the ground subsidence due to excess ice melt. This agrees with features observed at lake bottoms in the study area (Supplementary Fig. 4c, d).

Yedoma deposits, which in their undegraded state have a negligible microtopography (Fig. 2c) and contain the largest amounts of excess ground ice (Table 1), showed the most pronounced changes in the landscape configuration. Here, the timing of initial and advanced degradation was mainly dependent on the hydrological conditions (Fig. 2a). Under water-logged conditions, initial degradation occurred a few years after the start of the simulation, and water bodies formed after two to three decades of simulation time. Irrespective of the warming scenario, the landscape turned into a water body by the end of the twenty-first century, but a talik formed only under RCP8.5 warming (Supplementary Fig. 5l, o). Under well-drained conditions, initial degradation occurred after about two decades, and HCPs evolved after six to seven decades under both warming scenarios. Under RCP8.5, the final state of the landscape was characterised by massive subsidence of polygon troughs and rims, leaving a pronounced high-centred relief. This is reminiscent of conical thermokarst mounds (termed baidzharakhs), which can already today be observed at local Yedoma exposures and on top of Yedoma deposits[7] (Supplementary Figs. 4b, 5i).

Under variation of ground ice contents and hydrological conditions, our model can reproduce a multitude of degradation pathways for ice-rich permafrost landscapes underlain by ice wedges, which have recently been observed as widespread degradation features at other sites in the continuous permafrost zone[12,40,41]. Within the NESAL, such thaw phenomena do occur

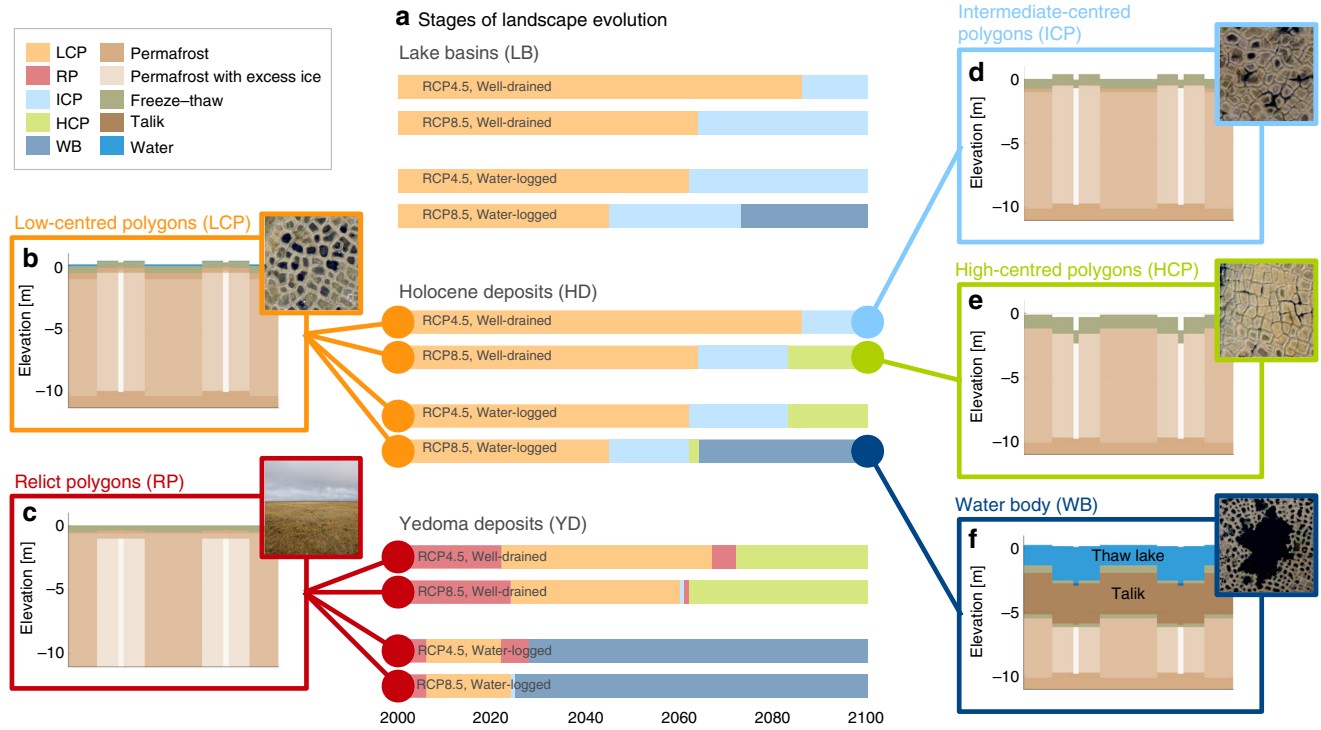

**Fig. 2 Simulated evolution of ice-rich permafrost landscapes.** Active low-centred polygons of drained lake basins and Holocene deposits (LCP, **b**), as well as relict polygons of Yedoma deposits (RP, **c**), initially host undegraded ice wedges. Melting of ice wedges under warming climatic conditions causes subsidence of the ground and a change in the microtopographic state of the landscape (**a**). Depending on the strength of the warming (RCP4.5 versus RCP8.5) and the hydrological conditions (well-drained versus water-logged), the landscapes evolve into intermediate-centred polygons (ICP, **d**), high-centred polygons (HCP, **e**) or water bodies (WB, **f**) by the end of the twenty-first century. Insets show aerial images of different states of ice-wedge terrain from study sites in northeast Siberia (Samoylov Island and Kurungnakh Island in the central Lena River delta). The initial and final states of the landscapes as well as detailed trajectories of the soil surface altitudes are provided in Supplementary Figs. 3 and 5.

**Table 1 Surface and subsurface characteristics, areal coverage and number of soil samples of the different landscape types of the NESAL.**

| Parameter | Unit | Lake basins (LB) | Holocene deposits (HD) | Yedoma deposits (YD) |
|---|---|---|---|---|
| Initial topography | – | LCP | LCP | RP (flat) |
| Ice-wedge depth | m | 3.8 | 10 | 20 |
| Wedge-ice volume | % | 10 | 30 | 50 |
| Excess ice content | % | 13 | 19 | 25 |
| Ground ice content | % | 68 | 74 | 80 |
| Full stratigraphy | – | Supplementary Table 3 | Supplementary Table 4 | Supplementary Table 5 |
| Number of soil samples | – | 199 | 200 | 585 |
| Total area | km$^2$ | 140,000 | 290,000 | 63,000 |

locally[12], but are either restricted to extreme site-specific conditions or induced by disturbances of natural or anthropogenic origin. Examples include the accumulation of snow in topographic depressions[12,42], the removal of protective organic layers or vegetation[43], tundra and forest fires[44], or vehicle tracks. When introducing such conditions or disturbances into our model, the simulated timing of permafrost degradation markedly shifts to earlier years[37]—in agreement with observations[12]. Here, however, we concentrated on undisturbed initial conditions and evaluate the impact of climate warming.

The correspondence between simulated and observed degradation pathways, together with the comprehensive site-level comparison between simulated and observed permafrost characteristics conducted in a preceding work[37], build confidence in that our simulations represent true end-members of landscape evolution in the NESAL under projected twenty-first century

climate warming. We note that the processes represented in our model are tailored for its application on decadal-to-centennial time-scales. If the approach was applied to millenial time-scales, comprising also extended periods of colder climatic conditions, the accumulation of ground ice would need to be taken into account, as it counter-acts the melting of ice wedges[39]. Even though our numerical model does not explicitly incorporate meso-scale landscape features (e.g., thermo-erosional valleys) and their lateral interactions (e.g., drainage of thermokarst lakes), the simulations under contrasting hydrological conditions are reflective of a broad range of permafrost thaw dynamics at the microscale.

**Permafrost thaw and ground subsidence.** Next, we assess the degradation of permafrost that is associated with the dynamic

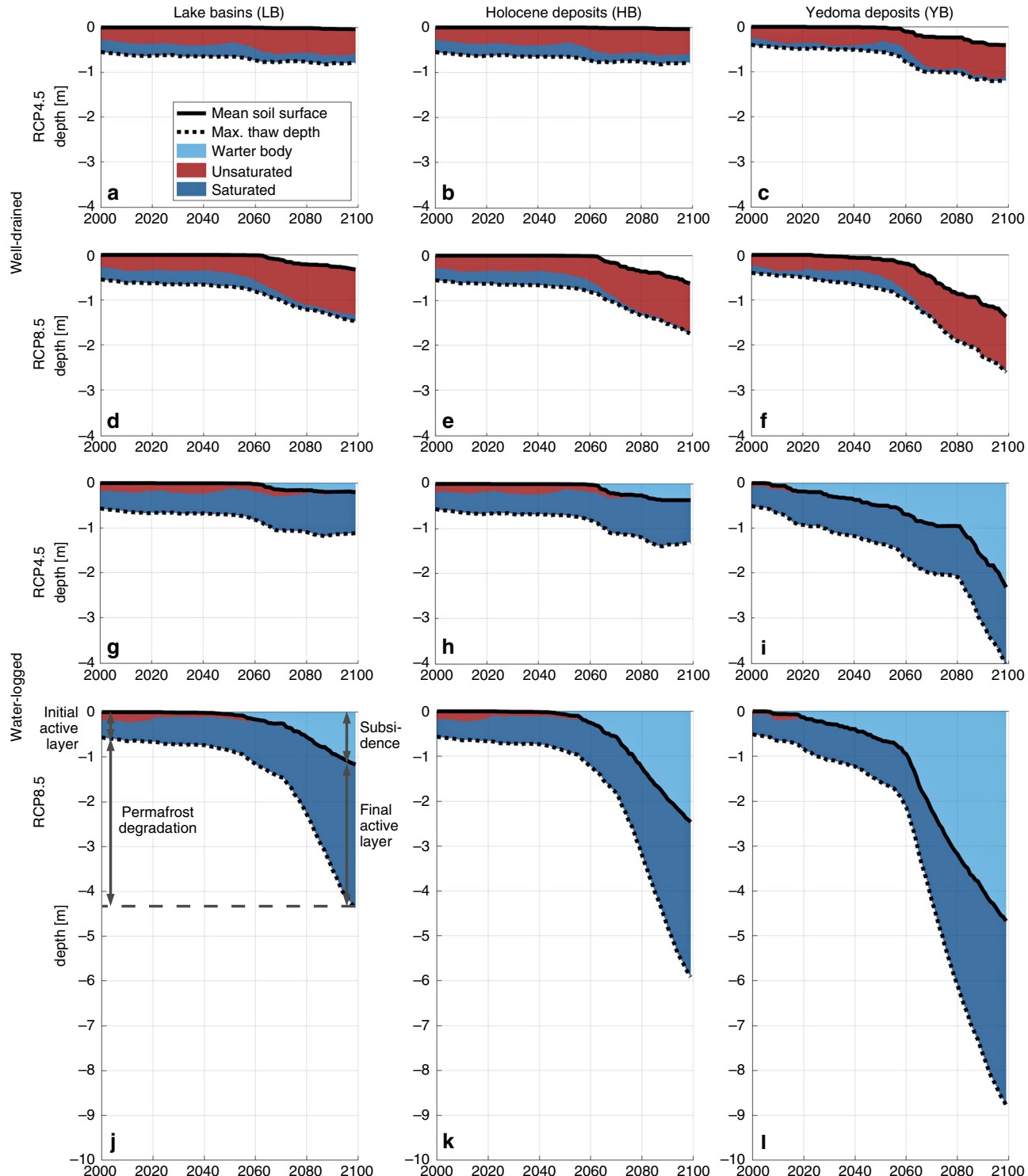

**Fig. 3 Simulated permafrost degradation and ground saturation.** The panels **a** to **l** show the accumulated mean ground subsidence and 11-year running mean of the maximum annual thaw depth for all settings (area-weighted means of the three tiles). Coloured areas indicate the fractions of unsaturated (red) and saturated (dark blue) conditions prevailing in the thawed ground throughout each year. Under water-logged conditions, surface water bodies form atop subsiding ground (light blue). Both ground subsidence and active-layer deepening cause permafrost degradation (see illustration in panel **j**). Corresponding plots for the reference runs without excess ice and the RCP2.6 runs are provided in Supplementary Figs. 6 and 7.

evolution of the different types of ice-wedge terrain. Figure 3 shows the temporal evolution of the maximum annual thaw depths as well as the mean ground subsidence (area-weighted means of the three tiles), for all landscape types and hydrological conditions under the RCP4.5 and RCP8.5 warming scenarios. Our numerical simulations provide an improved understanding

of the dynamics and controls of ice-rich permafrost thaw and allow a comparison with more simplistic model representations which only simulate gradual thawing of permafrost, ignoring thermokarst-inducing processes (Supplementary Methods 3 and Supplementary Fig. 6). Subsequently, we discuss four key results, which allow conclusions about the process of ice-rich permafrost

thaw in general, and about its relevance within the NESAL in particular.

First, substantial permafrost degradation was projected under the RCP4.5 and RCP8.5 warming scenarios, which is reflected in both increasing thaw depth relative to the soil surface, and ground subsidence resulting from excess ice melt. Simulated maximum thaw depths increased significantly within the twenty-first century (Fig. 3a–l), with relative increases ranging from factors of 1.3 (well-drained LB, RCP4.5) to 8.0 (water-logged YD, RCP8.5). Within the same period, thaw depths increased by factors of 1.7 (RCP4.5) and 2.3 (RCP8.5) in the corresponding reference runs, which do not represent thaw processes related to excess ground ice (Supplementary Fig. 6). These reference simulations do not reflect ground subsidence, which led to additional permafrost degradation of 0.2 m (well-drained LB, RCP4.5) to 4.7 m (water-logged YD, RCP8.5) by 2100 in the simulations with excess ice.

The degradation of ice wedges, which has so far only been observed locally within the study area[12], can be expected to occur as a widespread phenomenon across the cold permafrost of the NESAL within the twenty-first century, if warming exceeds RCP2.6 projections. Similar degradation of very cold, ice-rich permafrost has recently been reported for the High Canadian Arctic[41]. Both simulations and observations highlight the particular vulnerability of ice-rich permafrost landscapes to climate warming, which is despite very low permafrost temperatures at present[30]. It should be stressed that permafrost models without representation of excess ground ice can only simulate the gradual increase in thaw depth, but not the additional degradation through ground subsidence and associated feedbacks.

Second, after initial degradation, stabilisation of permafrost was projected towards the end of the twenty-first century under the moderate RCP4.5 warming scenario, but permafrost continued to degrade beyond 2100 under the strong RCP8.5 warming scenario. This qualitative difference between these two warming scenarios was found for all landscape types and hydrological conditions, except for water-logged Yedoma deposits. Under RCP4.5, the maximum thaw depths did not increase significantly during the final one to two decades of the simulations (Fig. 3a–c, g, h), while active-layer deepening and ground subsidence occurred during that period in all of the RCP8.5 simulations (Fig. 3d–f, j–l). Simulations for the ambitious mitigation scenario RCP2.6 revealed that ice-rich permafrost remained largely stable throughout the simulation period (Supplementary Methods 4 and Supplementary Fig. 7).

The stabilisation of the landscape under the warming scenarios depends on whether a sufficiently thick ice-poor layer consisting of thawed-out and laterally transported sediment accumulates. Such a layer would prevent the thaw front from reaching soil layers containing excess ice, i.e., a new active layer in equilibrium with the warming climate is established. On the one hand, the lateral transport of sediment (e.g., from polygon rims into deepening troughs) and the accumulation of sediment from melted excess ice layers promote the stabilisation of the landscape[39] (Supplementary Fig. 3). On the other hand, positive feedbacks induced by melting of excess ice cause an increase in thaw depths[39]. These include, for instance, soil warming resulting from increased snow depth in deepening troughs, and enhanced ground heat fluxes resulting from increased thermal conductivity of the active layer[37]. These results suggest that the possibility of permafrost stabilisation in the NESAL is linked to a critical threshold in the rate of climate warming reflected in the different warming scenarios. Under the strong and rapid warming of the RCP8.5 scenario, the system exceeds a tipping point beyond which stabilising feedbacks do not keep pace with positive feedbacks that accelerate permafrost degradation. Under the

moderate warming of the RCP4.5 scenario, negative feedbacks slow down permafrost degradation such that a new equilibrium active layer establishes towards the end of the simulation period. Our results thus emphasise the necessity of representing thermally stabilising and destabilising feedback processes in numerical permafrost models used to project the stability of near-surface permafrost in the future. Typically, these feedback processes involve lateral fluxes of mass and energy on spatial scales far below the grid size of current ESMs. Beyond this, our findings provide evidence that climate change mitigation according to the RCP2.6 or RCP4.5 scenarios could significantly limit the impacts of permafrost thaw on ecosystems and infrastructure in northeast Siberia, but most likely also in other Arctic regions hosting ice-rich permafrost.

Third, the abundance of excess ground ice exerts a strong control on the rate and magnitude of permafrost thaw. For instance, under well-drained conditions and RCP8.5 (Fig. 3d–f), the different landscape types showed a similar increase in maximum thaw depth during the simulation period, while the simulated subsidence increased from landscapes with low (LB) over intermediate (HD) to high (YD) excess ice contents. A similar relation was found under water-logged conditions and RCP8.5 (Fig. 3j–l), where subsidence is reflected in the deepening of thaw lakes forming during the simulation period. While a simple linear relation between total permafrost degradation and excess ground ice content can be established under the idealised assumptions of our numerical model (see Supplementary Notes 3 and Supplementary Fig. 8), the actual timing and the temporal evolution of permafrost thaw is further affected by site-specific factors, the incidence of extreme weather conditions, as well as hydrophysical and ecological feedback processes. In our simulations, this is illustrated by the formation of a deepening thaw lake for water-logged Yedoma deposits under RCP2.6 (Supplementary Fig. 7f) and RCP4.5 (Fig. 3i), which is not projected for landscapes with lower excess ice contents (Fig. 3g, h).

Our results underline the important role played by the present-day ground ice distribution, which is a product of climate-driven landscape evolution in the past, for permafrost thaw under a changing future climate. For landscapes with low abundance of wedge ice (e.g., drained lake basins), ground ice content has less influence on the magnitude and pace of permafrost degradation than in landscapes with high ice abundance (e.g., Yedoma deposits), where positive feedback mechanisms produce more rapid thawing, resulting in severe permafrost degradation and landscape collapse. Hence, robust projections of permafrost thaw require knowledge of present-day ground ice distribution[9] and numerical models that represent thermokarst-inducing processes in ice-rich terrain. Models lacking these processes likely underestimate permafrost thaw systematically. This is supported by the results of the reference runs with a simplistic representation of permafrost thaw dynamics (Supplementary Fig. 6), in which projected permafrost degradation was significantly lower compared with the simulations which include heterogeneously distributed excess ice.

Fourth, the hydrological regime of thawed ground tipped as a consequence of ice-wedge degradation to a dominance of either saturated or unsaturated conditions, depending on the specified hydrological conditions. At the beginning of the simulations, the fractions of saturated and unsaturated conditions in the thawed ground were of comparable magnitude, irrespective of whether the system is water-logged or well-drained. This was particularly the case for landscapes with undegraded ice-wedge polygons, i.e., pronounced LCP microtopography (LB and HD). Note that the presence of saturated ground conditions in the well-drained settings is possible, for instance, due to wet conditions after snowmelt and precipitation events, or due to higher water levels

in the depressed polygon centres, which are hydrologically isolated from the well-drained troughs. The subsidence of ground above melting ice wedges changes lateral water fluxes, mainly due to subsiding polygon rims no longer acting as barriers between polygon centres and inter-polygonal troughs. Under water-logged conditions, thaw subsidence led to inundation of the entire model domain, reflected in vanishing fractions of unsaturated conditions as soon as excess ice melt occurred (Fig. 3g–l). Under well-drained conditions, melting of ice wedges had a contrasting effect on the ground hydrological regime. Here, thaw subsidence of polygon rims improved the drainage of the landscape, leading to predominantly unsaturated conditions prevailing in thawed ground (Fig. 3a–f), particularly for those settings with emergent HCP microtopography (Fig. 3c, e, f).

These findings highlight the crucial role of ice-wedge thermokarst for the hydrological regime of the active layer and landscape hydrology. Melting of ice wedges and associated ground subsidence increase the lateral hydrological connectivity of the landscape[12], making the hydrological regime more sensitive to the surrounding conditions[37]. The water-logged cases correspond to situations where ice-wedge degradation leads to the development of deep water-filled troughs, and potentially the formation of thermokarst ponds and lakes. As soon as inter-polygonal troughs connect to an external drainage point, this can cause drainage of the entire landscape[12], leaving unsaturated high-centred polygons, drained troughs or conical thermokarst mounds (YD, Supplementary Fig. 4b). Overall, our results emphasise that the subsurface hydrology is crucially influenced by heterogeneous microtopography, lateral hydrological connectivity and ground subsidence. One-dimensional permafrost models that lack these complexities are thus inherently unsuitable for reliable projections of the hydrological regime of the active layer[21,45].

**Thaw-affected organic carbon stocks.** By scaling the simulated amounts of thawed organic carbon (per unit area) with the estimated total areas of each landscape type (LB, HD, YD) within the NESAL (see "Methods", Table 1), we estimated the proportion of the region's carbon pool that becomes subject to thawed conditions in the course of the twenty-first century under different warming scenarios (RCP2.6, RCP4.5 and RCP8.5; Fig. 4). Here, we distinguish between the simulations with excess ground ice and lateral fluxes, and the reference runs that reflect the typical representation of permafrost in ESMs. For the model runs with excess ground ice, the simulations under contrasting hydrological conditions (water-logged versus well-drained) provide a confining range for our estimates, acknowledging that the actual landscape evolution takes place under a range of different and dynamically changing hydrological conditions.

At the beginning of the twenty-first century ~7–8 Gt of organic carbon (GtC) were contained in the active layers of the simulated landscapes, and thus potentially accessible for microbial decomposition. Deepening of the active layers in combination with melting of excess ground ice caused thawing of additional carbon that was initially stored in perennially frozen soil layers. During the first half of the twenty-first century, the amounts of thawed organic carbon were projected to increase steadily, and the differences between the runs with excess ice and the reference simulations were small. By 2050, an additional 1.7 GtC (RCP2.6), 1.8 GtC (RCP4.5) and 2.5 GtC (RCP8.5), respectively, became subject to thawed conditions according to the reference runs. These numbers are within the respective ranges projected by the simulations with excess ice, reflecting that the increase was mainly attributable to active-layer deepening. However, during the second half of the twenty-first century excess ice melt set in under the RCP4.5 and RCP8.5 scenarios, leading to ground

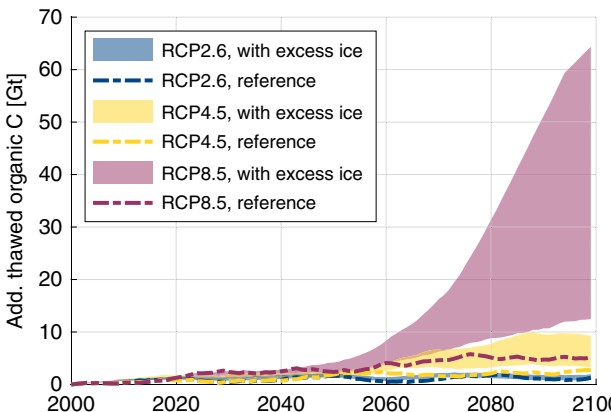

**Fig. 4 Projection of thaw-affected organic carbon stocks.** The estimates were obtained by scaling the simulation results using representative stratigraphies and the total areal extent of each landscape type within the northeast Siberian lowlands (see "Methods"). For each warming scenario, the indicated ranges correspond to 11-year running means of the annual maximum of thawed organic carbon under contrasting hydrological conditions. The reference runs without excess ice reflect the simplistic representation of permafrost in ESMs.

subsidence and an acceleration of permafrost thaw due to positive feedbacks (Fig. 3). Thereby, deep organic carbon stocks were subjected to thaw and would potentially become accessible for microbial decomposition. Hence, under the RCP4.5 and RCP8.5 scenario, after 2060 the projected range of thawed carbon was substantially higher in the runs with excess ice, compared with the respective reference runs. Under RCP4.5, by 2100 an additional 3.2–9.3 GtC were affected by thaw, if excess ice melt was taken into account, significantly exceeding the 2.7 GtC projected by the corresponding reference run. While stabilising feedback processes slowed down thawing rates under RCP4.5, positive feedbacks sustained rapid thawing under RCP8.5, exposing 12.5–64.4 GtC to thawed conditions by 2100, substantially exceeding the 5.3 GtC projected by the reference runs. Under the ambitious mitigation scenario RCP2.6, no widespread occurrence of excess ice melt was simulated (Supplementary Fig. 7), and consequently the amounts of carbon affected by thaw by 2100 (0.8–2.0 GtC) did not exceed the respective projection of the reference run (1.3 GtC). Overall, the deviation of the simulations with excess ice from the respective reference runs was found to increase with the strength of the warming scenario, reflecting that the contribution of thermokarst-inducing processes increased under stronger climate warming.

These results suggest that in regions like the NESAL which host cold, ice- and organic-rich permafrost deposits, substantially larger amounts of permafrost organic matter could thaw and expose carbon to mineralisation than what is projected by models like ESMs that employ a simplistic representation of permafrost thaw dynamics. Such global-scale models project significant amounts of thaw-affected permafrost carbon for the end of the twenty-first century (~140–400 GtC, depending on the scenario and the model[46,47]), but do not take into account thermokarst-inducing processes and deep carbon stocks which would become accessible through these processes. This is particularly problematic for cold permafrost regions like the NESAL where—even under RCP8.5—projected gradual thaw is limited (only ~5.3% of the NESAL's carbon pools become subject to thaw in our reference runs), but thermokarst-inducing process could cause thawing of 2–12 times (i.e., up to two-thirds of the NESAL's carbon pool) of the amount projected for gradual thaw only. ESMs might thus substantially underestimate the amounts of

carbon becoming accessible for microbial decomposition under a warming climate, particularly in cold and seemingly stable permafrost regions like the NESAL.

Our simulations further revealed that thermokarst-inducing processes are not only relevant for the deposits of the Yedoma domain which bear substantial amounts of well-preserved organic matter as well as relict wedge ice down to large depths[26] but also for landscapes hosting active ice-wedge polygons (LB and HD), where the same feedbacks cause rapid and deep thaw (Supplementary Fig. 8). Hence, in regions with ice-rich deposits prone to thermokarst processes[18], permafrost thaw through ice-wedge degradation needs to be taken into account in addition to gradual thaw through active-layer-deepening, when assessing the potential mobilisation of permafrost carbon pools under a warming climate. Since ice-wedge degradation constitutes a spacious process, it potentially affects larger areas and carbon stocks than localised or linear thermokarst features, such as retrogressive thaw slumps or coastal erosion. While these mass-wasting process might constitute an efficient pathway for the lateral export and the potential mobilisation of organic carbon[48], the total area and carbon pools affected by these processes are small[49,50] compared with the vast terrain underlain by ice wedges addressed in this study. We note that permafrost degradation beneath existing thermokarst lakes was not considered in this study, but constitutes another pathway of unlocking frozen carbon stocks that is typically ignored by ESMs[19,35].

Under RCP4.5 and RCP8.5, the spread between the simulations under contrasting hydrological conditions increased during the second half of the simulation period (Fig. 4), concurrent to the occurrence of ice-wedge degradation and associated ground subsidence (Fig. 3). This reflects that different pathways of ice-rich permafrost thaw under contrasting hydrological conditions affect substantially different amounts carbon, with water-logged conditions generally leading to deeper thaw than in well-drained settings (Supplementary Fig. 9). Moreover, the spread in thaw-affected carbon stocks was further found to increase with the strength of the warming scenarios. The spread in the projections can be interpreted as the uncertainty range of our simulations with respect to the preconditioning of the landscape hydrology and its response to a warming climate. In order to further constrain our estimates, more dedicated modelling studies are needed that recognise the spatial variability of the present-day landscapes' topography and hydrology (which have been affected by thermokarst and thermoerosion in the past), and at the same time realistically reflect the meso-scale hydrological and geomorphological responses to future permafrost degradation, such as the expansion and drainage of thermokarst lakes. Meso-scale snow redistribution (e.g., blowing snow from frozen lake surfaces) constitutes another process which can affect the thermal and hydrological state and thaw dynamics of permafrost. Constraining the hydrological response of Arctic lowlands to thawing of permafrost appears even more important when considering that lateral hydrological export might constitute also an important direct pathway for soil carbon loss[51].

Finally, the hydrological conditions do not only affect the total amounts of thawed organic carbon but also exert control over its potential decomposition pathways[11,52–54]. With advanced melting of excess ice, water-logged conditions will lead to a dominance of saturated active layers, and hence favour anaerobic decomposition pathways; well-drained conditions in turn will favour unsaturated soils, in which carbon decomposition occurs aerobically, especially if a high-centred microtopography with improved landscape drainage emerges[12] (Supplementary Fig. 9). The quantification of microbial decomposition, turnover into greenhouse gases, and potential gas fluxes into the atmosphere is, however, beyond the scope of this study, and would require the

extension of our physical process model with suitable biogeochemistry schemes. It should be stressed, that the overall carbon balance of permafrost ecosystems under a changing climate is depending on various processes besides permafrost thaw, including, for example, carbon uptake by vegetation. Our simulations do not allow conclusions about which portion of the thaw-affected carbon stocks presented in Fig. 4 could end up as greenhouse gases in the atmosphere, and whether the study region would evolve into a carbon source or sink.

In conclusion, the process-based simulations presented in this study provide evidence for a substantial potential for unlocking of vast amounts of currently frozen organic carbon pools in cold ice-rich lowlands, through thermokarst-related permafrost thaw in response to a warming climate. Thermokarst-inducing processes merit representation in ESMs as they could significantly contribute to the global permafrost carbon–climate feedback[11,23] already during the twenty-first century, and also at lower warming levels than previously thought[55]. According to our results, mitigation of climate change could save ecosystems in northeast Siberia from severe permafrost degradation and landscape collapse that are likely to occur under a high emissions scenario (RCP8.5).

## Methods

**Numerical model of ice-wedge terrain**. We used a version of the CryoGrid 3 land surface model[13] that represents the surface topography of ice-wedge terrain using three tiles[37]: polygon centres, polygon rims and inter-polygonal troughs. Each tile has a one-dimensional vertical representation of the subsurface, for which the thermal and hydrological dynamics were simulated by solving the heat-conduction equation with phase change, combined with a hydrology scheme for unfrozen ground[37,56]. The snow scheme simulated the dynamic build-up and ablation of the snow pack as well as infiltration and refreezing of rain and meltwater. The model takes into account lateral fluxes of heat, water and snow between the tiles, assuming a circular symmetry for individual ice-wedge polygons (see Fig. 5; Supplementary Methods 2, Supplementary Fig. 10 and Supplementary Table 1). Lateral drainage of water from the troughs into an external reservoir is also possible. The elevation of this reservoir ($e_{res}$) was varied between different model runs in order to prescribe different hydrological conditions.

The geomorphological evolution of ice-wedge terrain was represented in our model by the combination of two process. First, we used the excess ice scheme introduced by Lee et al.[10], which was implemented into CryoGrid[13,35] to simulate the subsidence of the soil surface resulting from melting of excess ground ice. Soil layers with ice contents above the natural porosity were considered to contain excess ice. Excess water is moved upwards upon thawing of soil layers containing excess ice, while mineral and organic sediment is routed downwards. This scheme reflects the main process causing the degradation of ice-wedge polygons[37]. Second, we used the non-linear hillslope diffusion scheme by Roering et al.[57], which has been adapted for periglacial environments in previous studies[58,59]. This scheme describes the lateral erosion of mineral and organic sediment, which is an important process promoting the stabilisation of ice-wedge polygons[39].

For this study, we introduced the lateral sediment transport scheme into CryoGrid 3, by prescribing a slope-dependent lateral sediment flux $q^{sed}_{i \leftarrow j}$ [m s$^{-1}$] between two adjacent tiles $i$ and $j$. We neglected diffusive contributions to $q^{sed}$, and restricted the lateral sediment transport to advective fluxes, which occur after rapid ground subsidence that gives rise to steep terrain gradients. Advective fluxes were calculated as follows (see Supplementary Methods 2 for details):

$$q^{adv}_{i \leftarrow j} = K_{eff} \frac{a_j - a_i}{D_{ij}} \frac{\alpha^2}{\alpha^2_{crit} - \alpha^2} \frac{L_{ij}}{A_i} \tag{1}$$

where $a_{i,j}$ is the soil surface altitude of the tiles, $\alpha = \arctan\left(\frac{a_j - a_i}{D_{ij}}\right)$ is the angle of the slope between the tiles and $\alpha_{crit}$ a critical slope angle at which advective fluxes diverge (Supplementary Fig. 11). The topological relations among the tiles involved are reflected by their respective distances ($D_{ij}$), contact lengths ($L_{ij}$) and areas ($A_i$). $K_{eff}$ [m$^2$ s$^{-1}$] is a transport coefficient composed of a subaerial ($K_{land}$) and a subaqueous ($K_{water}$) part, depending on the surface conditions of both tiles (Supplementary Fig. 12). We set $\alpha_{crit} = 45°$, $K_{land} = 3 \times 10^{-10}$ m$^2$ s$^{-1}$, and $K_{water} = 3 \times 10^{-8}$ m$^2$ s$^{-1}$, following previous studies in periglacial environments[58,59] (Supplementary Table 2).

**Ground stratigraphies and further parameters**. The three different landscape types (Fig. 1; two from the Yedoma region including Late Pleistocene Yedoma deposits (YD) and drained lake basins (LB), and Holocene Deposits (HD)) differ in

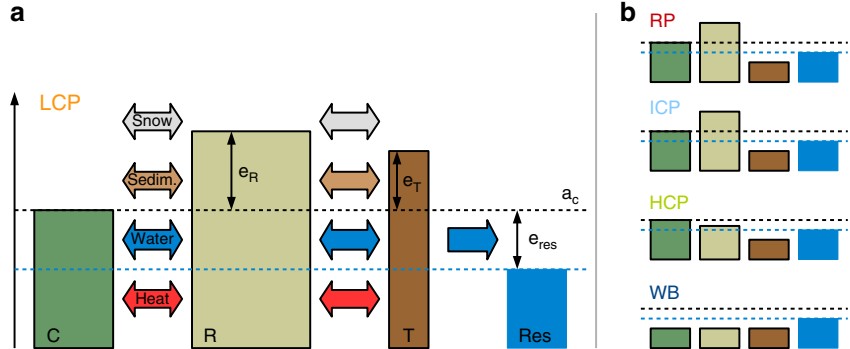

**Fig. 5 Schematic of the model set-up. a** Coupled tiles (polygon centres (C), polygon rims (R) and troughs (T)) and an external water reservoir (res) are used to represent surface and subsurface heterogeneities of ice-wedge terrain. The surface microtopography is reflected in different initial elevations (e) relative to the altitude of the centres ($a_C$). Different subsurface stratigraphies are provided in Supplementary Tables 3–5. Lateral fluxes of heat, water, snow and sediment between the tiles were calculated taking into account topological relationships among the tiles (see Supplementary Methods 2 and Supplementary Fig. 10 for details). The parameter $e_{res}$ was used to specify the hydrological conditions. **b** The microtopographic state of the landscape is defined according to Eqs. (2) to (6) as relict polygons (RP), low-centred polygons (LCP), intermediate-centred polygons (ICP), high-centred polygons (HCP) or water body (WB), depending on the soil surface altitudes of the tiles ($a_{C,R,T}$) and the elevation of the external water reservoir ($e_{res}$).

their surface and subsurface characteristics. For LB and HD, which are characterised by active ice-wedge polygons, we assumed a low-centred surface microtopography ($e_R = 0.4$ m, $e_T = 0.3$ m). Undegraded YD, which are underlain by relict ice-wedge polygons, do not show a pronounced surface microtopography, so we assumed a flat surface ($e_R = 0.0$ m, $e_T = 0.0$ m).

The subsurface ground ice distributions of LB, HD and YD were reflected by setting up the model with different cryostratigraphies. For this, we used representative values[25,60] for wedge-ice volumes, ice-wedge depths and ground ice contents (Table 1). While we assumed homogeneous ice-wedge dimensions with depth for HD and YD, the pronounced wedge-shape of ice wedges in LB was reflected by reducing ground ice contents of the trough tile in deeper soil layers. An ice-rich intermediate layer[39] of 0.2 m (LB, HD) to 0.4 m (YD) thickness was placed above the ice wedges of all types. Organic, mineral and ice contents of LB, HD and YD were based on a total of 984 soil samples from different sites of all landscape types within the NESAL (see Table 1 for the number of samples per landscape type); the samples were measured with carbon–nitrogen (elementar vario EL III) and total organic carbon (elementar vario MAX C) analysers. Based on these data, we specified the soil stratigraphies in the model which we consider to be representative for the respective landscape type within the entire study area. Detailed stratigraphies are provided in Supplementary Tables 3–5.

In agreement with observations[29], the density of snow at deposition was set to $\rho_{snow} = 250$ kg m$^{-3}$, and the natural porosity of soil layers containing excess ice was set to $\phi_{nat} = 0.55$. All other model parameter values are provided in the Supplementary Information, or were set to the same values as in previous studies for the same study area[13,37].

**Forcing data**. We used meteorological forcing data for the central Lena River delta; these data have been used in previous studies using CryoGrid 3[13,35]. The forcing data covered the period from 1901 until 2100, and were composed of downscaled CRU-NCEP v5.3 data for the 1901 to 2014 period, and additionally applied anomalies from CCSM4 projections under the RCP4.5 and RCP8.5 scenarios (see ref. [13]. for details). For the RCP2.6 scenario, we took anomalies from CCSM4 projections and applied them repeatedly to the period from 2000 until 2014 of the RCP4.5 forcing data. Due to the central location of the Lena River delta within the NESAL, we assumed the current climate characteristics and future climate trends throughout the NESAL to be comparable with those expected for the Lena River delta. Differences in current climate characteristics within the NESAL were assumed to be negligible compared with the differences expected from the climate warming scenarios.

**Simulations and spin-up**. For each landscape type (LB, HD and YD), we conducted simulations under three climatic forcing scenarios (RCP2.6, RCP4.5 and RCP8.5) and two contrasting hydrological conditions (water-logged, well-drained). The two contrasting hydrological conditions were prescribed through fixed elevations of the external water reservoir ($e_{res}$), which was connected to the trough tile (Fig. 5). We set $e_{res} = 0.0$ m (LB, HD) or $e_{res} = -0.2$ m (YD) to reflect water-logged, and $e_{res} = -10.0$ m to reflect well-drained hydrological conditions. The initial temperature profiles of all 12 simulations were based on the borehole data, and a subsequent 50-year spin-up period using historical forcing data (10/1949-12/1999). The analysed period of the simulations was from 01/2000 until 12/2099.

In addition to the runs with the tiled set-up, we conducted reference runs under the three climatic forcing scenarios (RCP2.6, RCP4.5 and RCP8.5) with a setting of CryoGrid 3, which is comparable with the design of ESM land surface schemes.

These runs were conducted using a one-dimensional subsurface representation without excess ice (Supplementary Table 6) and without taking into account lateral fluxes (see Supplementary Methods 3 for details).

**Model evaluation and sensitivity analysis**. The model set-up for ice-wedge terrain was evaluated against field measurements from a study site in the central Lena River delta[29] in a preceding study[37]. The study showed that simulated soil temperatures, thaw depths, soil moisture levels, water tables, snow heights and surface energy fluxes for different microtopographic features of ice-wedge terrain (polygon centres and rims) agreed well with observations. The correspondence between simulated and observed degradation pathways and landforms provides further evidence for the validity of our modelling approach (see main text, Fig. 2 and Supplementary Fig. 4).

In order to ensure the robustness of our results and to identify critical processes, we conducted a sensitivity analysis in which single-model parameters deviated from their default values. The parameters were related to different processes, and included the fresh snow density ($\rho_{snow}$), the sediment transport coefficients ($K_{land}$, $K_{water}$), the field capacity ($\theta_{fc}$) and the natural porosity of ice-rich soil layers ($\phi_{nat}$). Results of the sensitivity analysis are provided in Supplementary Fig. 13, and discussed in the Supplementary Notes 4.

**Model diagnostics**. During the simulations the topography of the landscape was altered due to ground subsidence resulting from excess ice melt and loss, as well as lateral sediment fluxes between adjacent tiles. While ground subsidence is indicative of permafrost degradation, lateral sediment fluxes smooth out the topography and potentially stabilise low-lying tiles. This process can counteract ice-wedge degradation[39]. Depending on the soil surface altitude of the three tiles ($a_{C,R,T}$) and the altitude of the external water reservoir ($a_{res} = a_C + e_{res}$), we distinguished five different states of the surface microtopography (Fig. 5), each corresponding to different landforms:

$$\text{Relict polygons (RP:)} \quad a_T = a_C = a_R, \tag{2}$$

$$\text{Low-centred polygons (LCP):} \quad a_C < \min(a_R, a_T), \tag{3}$$

$$\text{Intermediate-centred polygons (ICP):} \quad a_T < a_C < a_R, \tag{4}$$

$$\text{High-centred polygons (HCP):} \quad a_C > \max(a_R, a_T), \tag{5}$$

$$\text{Water body (WB):} \quad \max(a_C, a_R, a_T) < a_{res}. \tag{6}$$

Maximum thaw depths are the area-weighted mean of the maximum thaw depth of each tile for each year. The proportion of unsaturated conditions corresponds to the fraction of thawed soil cells (above the permafrost) throughout each year that contain a non-zero fraction of air; the remaining fraction of thawed cells is saturated. Note that this definition takes into account changing thaw depths and changing water table levels throughout the year.

**Areal extent and carbon stocks of the NESAL**. Our overall region of interest is the northeast Siberian Arctic, defined as all landmass north of 66°N within the political borders of Yakutia (~1,462,000 km$^2$; Fig. 1a; Supplementary Fig. 2). Within this region, we identified three types of ice-rich lowlands underlain by ice wedges (LB, HD and YD), and the union of these constitute the northeast Siberian

Arctic lowlands (NESAL). Based on different sources[25,26,61], drained lake basins (LB, ~140,000 km$^2$) were assumed to comprise 69% of the mapped Yedoma region[62] (excluding existing lakes), while the remaining 31% of were considered as undisturbed Yedoma deposits (YD, ~63,000 km$^2$; Table 1). The area of lowlands with ice-rich deposits outside the Yedoma region (Holocene deposits, HD) was estimated based on a digital elevation model (DEM), which had a spatial resolution of 3 arc seconds[63]. We assumed ice-wedge polygonal tundra to be present in terrain with an altitude between 5 and 100 m above sea level, and a slope ≤ 4°. This region covers ~290,000 km$^2$ of the northeast Siberian Arctic. By using these criteria, we excluded low-lying terrain like floodplains, as well as slopes and mountain regions. The areal coverage of HD was validated using another DEM compiled from TanDEM-X data[64], yielding a relative difference in the HD area of < 3%. We excluded large water bodies (old thaw lakes, larger rivers) from the area estimates for all landscape types. In total, the NESAL cover about 493,000 km$^2$ (34%) of the overall study region. Note that the NESAL region mapped according to the criteria above largely falls into regions with high or very high thermokarst landscape coverage of types Lake and Hillslope mapped in an expert assessment[18].

The organic carbon contents assumed for the ground stratigraphies of the different landscape types in the simulations were based on a large number of soil samples from several study sites within the NESAL (Table 1). The ground stratigraphies were thus considered to be representative for the entire NESAL. We estimated the NESAL's thaw-affected carbon pool by scaling the amounts of organic carbon per unit area simulated for each landscape type at a single location with the total areas of the respective landscape types (LB, HD and YD) within the NESAL (Table 1).

## Data availability

The data that support the findings of this study are available from the corresponding author upon reasonable request.

## Code availability

The model code and the settings to reproduce the numerical simulations are available from https://doi.org/10.5281/zenodo.3648266. The code is published under a GNU General Public License v3.0.

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

## Acknowledgements

The authors gratefully acknowledge the Climate Geography Group at the Humboldt University of Berlin for providing resources on their high-performance-computer system. We thank Lutz Schirrmeister for providing soil organic carbon data and Alexander Oehme for processing the RCP2.6 forcing data. This work was supported by a grant of the Research Council of Norway (project PERMANOR, grant no. 255331). J.N. was supported by a grant of the German Academic Exchange Service (DAAD) for a research stay at the University of Oslo, and by the Geo.X Research Network. S.W. acknowledges funding through Nunataryuk (EU grant agreement no. 773421). M.L. was supported by a BMBF grant (project PermaRisk, grant no. 01LN1709A). J.S. was supported by a NERC-BMBF grant (project CACOON, grant no. 03F0806A) and the International Permafrost Association (Action Group The Yedoma Region). This work was supported by funding from the Helmholtz Association in the framework of MOSES (Modular Observation Solutions for Earth Systems).

## Author contributions

J.N. designed the study, developed and implemented the numerical model, carried out and analysed the simulations, prepared the results figures and led the paper preparation. S.W., M.L. and J.B. co-designed the study, and interpreted the results. L.M. contributed to the development and implementation of the numerical model, and prepared schematic figures. J.S. contributed soil organic carbon and ice-wedge data, field observations and set the selection criteria for landscape-type mapping. S.L. did GIS analysis and mapping, and prepared geospatial figures. All authors contributed to the writing and editing of the paper.

## Competing interests

The authors declare no competing interests.

## Additional information

**Peer review infromation** *Nature Communications* thanks Merritt Turetsky and other, anonymous, reviewers for their contributions to the peer review of this work. Peer review reports are available.

