## [Peer Review File · Nature Communications]

Reviewers' comments:

Reviewer #1 (Remarks to the Author):

Review of Nitzbon et al.

This manuscript describes a numerical model that represents thermokarst processes in ice-rich terrain common to northern Siberian lowlands. The model focuses on several thermokarst processes present in the study region, with a particular focus on ice wedge dynamics. This study is timely, particularly in light of the recent IPCC special report on oceans and the cryosphere. While an increasing number of Earth system models are beginning to incorporate permafrost into their frameworks, few if any address permafrost thaw initiated by thermokarst or thermoerosion.

Overall this paper is well written, the approach is described well, and the results are interesting. It is clear that this modeling approach is an advancement over models that cannot simulate change in ice-rich permafrost, including land surface subsidence. The style of the writing is quite dense – while all interesting it was difficult at times to discern the key findings that the authors wished to emphasize over others. The results largely focus on hydrophysical changes to the landscape and surface expressions that result from the thermokarst simulations. The paper also uses soil and sediment data from the region to estimate the stock of permafrost carbon exposed to thawed conditions. This is the weakest portion of the paper, and uses inappropriate terminology in several locations.

This paper will be an important contribution to the permafrost and modeling communities but given the detail and complexity of the simulations, might be best received in a more disciplinary journal. I would stress, however, that a number of papers addressing thermokarst have been published in high profile journals (including Nature journals) in recent years, but these tend to focus on estimates of permafrost carbon release. This paper does not address carbon release associated with permafrost thaw, but does provide a new modeling framework that could be an important first step towards this goal.

While there is a lot of detail and very nice graphics presented in this manuscript, the text raises a number of important issues and questions that should be addressed before this manuscript can be considered ready for publication:

- 1) While the manuscript clearly describes (and visualizes!) which types of thermokarst and surface expression change are included in the simulations, there is no clear sense of whether these are the dominant types of thaw/thermokarst in the study region. Can the authors use remote sensing or other spatial data to provide an assessment of whether they are representing the dominant pathways ice-rich permafrost thaw and resulting landscape change? What types of thaw/thermokarst are missing and how important are these trajectories in the study area?
- 2) It is unclear to me whether existing (old) thaw lakes or alas are included in the simulations. These are major features on the landscape, presumably are important to water and nutrient flow, and certainly need to be taken into account when estimating potential carbon mobilization.
- 3) Walter Anthony et al. (2018) found that RCP8.5 warming affected yedoma carbon release very differently than RCP4.5 warming because of more rapid thaw lake drainage under RCP8.5. This is a mechanism by which thaw lake carbon mobilization could be greater under mitigation scenarios relative to RCP8.5 warming scenarios. It is not clear to me whether this study is able to address this issue. If not, how much of the results of this study are likely to change if thaw lake drainage is included more explicitly?
- 4) The authors present simulation results that in general conform to other modeling studies, but do not compare results from the model to any empirical information. I am not an expert in modeling

myself so do not have an expert opinion on what would be considered standard. But it seems important to evaluate the model performance more fully than what is presented in the current manuscript.

5) The authors need to use more appropriate terminology in the carbon section. The manuscript currently does a rough calculation of how much permafrost carbon (i.e., a C stock assessment) will be exposed to thaw under different scenarios. This really has nothing to do with mobilization and it is misleading for the authors to use terms like that. In my opinion, this entire section needs to be carefully re-thought.

6) The authors make a number of simulations in their modeling framework and simulations, such as decisions regarding aerobic: anaerobic proportions. It seems important for this manuscript to provide an assessment of the sensitivity of model performance to these assumptions and to the suite of parameters involved in the simulations.

7) As a more minor point, I do not think that the term "conservation" is appropriate for use in the title of this manuscript. To most scientists, conservation implies active protection or management of the land or resource. What the authors are really hinting at in the title is whether response of ice-rich permafrost will be a slow or fast response, relative to change of ice-poor permafrost.

Reviewer #2 (Remarks to the Author):

Major comments

Developing a process-based understanding of polygonal tundra evolution and linking it to the permafrost carbon feedback is important piece of the permafrost puzzle that is currently missing from the global scale permafrost modeling picture. This work is timely and addressing one of the pressing issues of the Arctic, the fate of carbon from the organic rich frozen ground of the tundra type ecosystem. Below I have several questions that I hope authors will take in the positive spirit intended during revision of this important paper.

1. The effective hillslope sediment transport model was adapted to tundra environment to model polygonal tundra evolution. Why did authors decide to adapt hillslope sediment transport model instead of adapting one of the existing erosion model by the corresponding rescaling of geological years?

2. Currently, model simulates progressive changes leading transition from one state to another with the corresponding permafrost degradation. From field experiments, we know that LCP can also have small ice wedges in the middle (e.g. studies by Jastrow et al), suggesting changes from HCP to LCP and back. However, it is not clear under what conditions it is possible to cyclically develop a dynamical evolution of HCP to LCP and vice versa? Applying the RCP 2.6. scenario might be useful to illustrate how negative trend in the global temperatures during this century could affect changes in polygonal tundra landscapes. It also could be interesting to show that these transitions are possible. For example, Kleinen and Brovkin (2018) drew some insightful conclusions based RCP 2.6 suggesting that under this scenario almost double amount of carbon need to be removed from the atmosphere in order to reach the 1.5oC target.

3. Finally, authors used simplified empirical relationship to address the amount carbon that could be released as a result of permafrost thawing. Recent studies show that observation from flux towers indicate that most of this release is not captured by the near-by flux towers, suggesting significant lateral transport of the carbon in the form of the dissolved organic carbon (Plaza et al. 2019). Taking this into account how it might change the current numbers on Figure 4.

Minor comments

Authors claim that there is such a strong dependence between subsidence, amount of ground ice, and carbon emissions. I suggest to include plots illustrating changes in the carbon emission with respect to subsidence and ground ice. This might to conceptualize these relationships and benefit future studies working on similar topic.

From page 8. "Locally observed degradation" It is not clear does the model address the effect of the anthropogenic origins or not?

It would be nice to know the percent difference between projected carbon release by global models from Russian permafrost and current study only for tundra over 21st century. Showing this difference can make a greater impact and better convince readers about importance of the tundra (yedoma) permafrost thaw.

References

Kleinen T and Brovkin V 2018 Pathway-dependent fate of permafrost region carbon Environ. Res. Lett. 13 094001
Plaza C, Pegoraro E, Bracho R, Celis G, Crummer KG, Hutchings JA, Pries CEH, Mauritz M, Natali SM, Salmon VG, Schädel C, Webb EE, Schuur EAG (2019) Direct observation of permafrost degradation and rapid soil carbon loss in tundra. Nat Geosci 12:627–631. <https://doi.org/10.1038/s41561-019-0387-6>

Please find below responses to the issues raised by the two reviewers. Reviewer comments are in **boldface** and extracts from the manuscript are in *italics*, changes to the manuscript are highlighted **yellow**. All references used in the responses below are either contained in the reference list of the revised manuscript or provided at the end of this response. We also provide a change-highlighted version of the revised manuscript where all changes compared to the initial submission are visualized.

Reviewer #1:

We thank the reviewer for the thorough evaluation of our manuscript and for pointing out weak points which deserve improvement. We address the critiques raised by the reviewer point-by-point below:

[...]

The style of the writing is quite dense – while all interesting it was difficult at times to discern the key findings that the authors wished to emphasize over others.

We agree with this point which was similarly raised by the editor. We emphasize our rationale and the key results more clearly in the revised version of the manuscript. To give one example, we revised the introduction section to work out the objectives more clearly and complemented it by a final paragraph, summarizing the key findings and conclusions:

We demonstrate that our numerical modelling approach can retrace a multitude of landscape evolution and degradation pathways characteristic to ice-wedge terrain. Projected permafrost degradation in the NESAL during the twenty-first century is substantially increased when ground subsidence due to melting of ground ice and small-scale lateral fluxes are taken into account. The response of ice-rich permafrost terrain differs considerably for different future warming scenarios, ranging from mostly stable landscapes (RCP2.6) to widespread landscape collapse (RCP8.5). We find that lateral sediment transport moderates initial permafrost degradation, leading to stabilized landscapes by 2100 under RCP4.5. For RCP8.5, we find that substantial amounts of the NESAL's organic carbon stocks might become subject to thaw within the present century, despite the projected stability of the region in previous ESM projections. This study emphasizes the necessity of representing thaw pathways characteristic for ice-rich permafrost in ESMs and provides significant progress towards achieving this goal. More broadly, our results underline the importance of mitigating climate change if we are to limit its impacts on permafrost ecosystems.

We further clarified several statements in the “Results and Discussion” section which are too many to be presented here. Specific changes are visualized in the “change-highlighted” version of the manuscript.

[...]

1) While the manuscript clearly describes (and visualizes!) which types of thermokarst and surface expression change are included in the simulations, there is no clear sense of whether these are the dominant types of thaw/thermokarst in the study region. Can the authors use remote sensing or other spatial data to provide an assessment of whether they are representing the dominant pathways ice-rich permafrost thaw and resulting landscape change?

We restricted our analysis to terrain underlain by ice wedges which is the dominant type of massive ground ice in the lowlands of northeast Siberia (e.g., Fedorov et al. (2018)). Other types of massive ground ice such as buried glacier ice, buried lake ice, or buried snowpacks are not common in the study area (e.g., Schirrmeyer et al. (2008)). We argue that when ice-rich permafrost thaws, wedge ice is involved at the micro-scale, i.e. the scale of individual ice-wedge polygons. At the meso-scale (or “landscape-scale”) spacious degradation of ice wedges can lead to the emergence of larger-scale thaw features, such as thermo-erosional valleys or thermokarst lakes. While the spatio-temporal dynamics of these features (e.g., the expansion or drainage of thermokarst lakes) is not explicitly modelled, the range of involved thaw processes which take place at the micro-scale are reflected by our simulations under contrasting hydrological conditions (water-logged versus well-drained). In the revised manuscript, we explained this approach in the introduction:

Our focus here is on vast lowlands (covering about 493,000 km²) which are underlain by massive ice wedges in the subsurface. Wedge ice constitutes the dominant type of ground ice in the NESAL [Schirrmeyer et al. (2008), Fedorov et al. (2018)] and consequently most rapid permafrost thaw features are assumed to involve the degradation of ice wedges at a micro-scale. Acknowledging the landscape evolution throughout the Holocene (Supplementary Figure 1), we distinguish three major types of present-day landscapes, with marked differences in the wedge ice volumes and the thickness of ice-rich deposits (Figure 1 b): drained thermokarst lake basins (LB), undisturbed Holocene deposits (HD), and undisturbed Yedoma deposits (YD).

We visually compared the extent of the mapped lowlands in northeast Siberia (NESAL) to the map of thermokarst landscape coverage by Olefeldt et al. (2016) and found a good agreement with regions that were classified to have “high” or “very high” thermokarst landscape coverage of the types “Lake” and “Hillslope”. We mention this in the Methods section:

Note that the NESAL region mapped according to the criteria above largely falls into regions with “high” or “very high” thermokarst landscape coverage of types “Lake” and “Hillslope” mapped in an expert assessment [Olefeldt et al. (2016)].

To allow for a better impression of the extent of the landscapes addressed in our study, we added the map of the study region which is also provided in the supplement, to Figure 1 (panel a).

What types of thaw/thermokarst are missing and how important are these trajectories in the study area?

By restricting our study to lowland areas, we a priori exclude some rapid thaw processes occurring only on hillslopes such as landslides or active layer detachments. Our study furthermore does not address some types of rapid thaw processes related to ice-rich permafrost lowlands, which we argue to be of minor importance in the study area relative to spacious ice-wedge degradation that affects an estimated 493,000 km². These include permafrost degradation through mass-wasting lateral erosion processes such as retrogressive thaw slumps or coastal erosion. Nitze et al. (2018) estimated that between 1999 and 2014 a total area of 1.08 km² was affected by 140 retrogressive thaw slumps within a transect of about 43,000 km² which intersects with a large part of our study area. Even if thaw slumps would become more abundant in a warming climate, the total area directly affected would still be small compared to our estimated areas of ice-wedge terrain. Similarly, Günther et al. (2013) estimated recent erosion rates of ice-rich coastline along the Laptev sea (about 1400 km) to be in the order of 5 m/yr. Assuming a more pessimistic value of 10 m/yr over a 100 year

time-frame, would yield a total affected area of 1,400 km². The corresponding amount of eroded organic carbon (OC) was estimated to 0.7 Tg/yr. Assuming a value of 1 Tg/yr over a 100 year time-frame would yield a total OC amount of 0.1 Pg (=Gt).

We are thus confident that the overall areas directly affected by mass-wasting lateral erosion processes are small compared to the total areas affected by spacious ice-wedge degradation. However, as the export and mobilization of thawed OC from lateral mass-wasting processes might be more efficient compared to flat inland terrain (e.g., Tanski et al. (2019)), these processes also need to be taken into account in comprehensive assessments of carbon mobilization from thawing permafrost. This is, however, beyond the scope of our study. In the revised version of our manuscript we add a discussion of these processes:

Since ice-wedge degradation constitutes a spacious process, it potentially affects larger areas and carbon stocks than localized or linear thermokarst features such as retrogressive thaw slumps or coastal erosion. While these mass-wasting process might constitute an efficient pathway for the lateral export and the potential mobilization of organic carbon [Tanski et al. (2019)], the total area and carbon pools affected by these processes are small [Günther et al. (2013), Nitze et al. (2018)] compared to the vast terrain underlain by ice wedges addressed in this study.

We further mention that meso-scale thermokarst features are not explicitly modelled:

Even though our numerical model does not explicitly incorporate meso-scale landscape features (e.g., thermo-erosional valleys) and their lateral interactions (e.g., drainage of thermokarst lakes), the simulations under contrasting hydrological conditions are reflective of a broad range of permafrost thaw dynamics at the micro-scale.

The representation of already existing old thermokarst lakes is further explained in the response to the next point.

2) It is unclear to me whether existing (old) thaw lakes or alas are included in the simulations. These are major features on the landscape, presumably are important to water and nutrient flow, and certainly need to be taken into account when estimating potential carbon mobilization.

Presently existing thaw lakes were not explicitly included in our study, and the area covered with thaw lakes was excluded from the upscaling of thawed OC. We agree that these lakes constitute an important factor in terms of carbon mobilization, but our study puts the focus on terrestrial terrain which is currently underlain by ice wedges. However, our simulations do reflect the conversion of ice-wedge terrain into new thaw lakes (see for instance Fig. 2 a,f). Drained thaw lakes (“alas”) are explicitly included (see Fig. 1b “LB”) and were assumed to make up 69% of the Yedoma region and to feature rather shallow ice wedges with a low wedge-ice volume, following Strauss et al. (2013) and Ulrich et al. (2014). Alas outside the Yedoma region would be considered to be of the Holocene deposit type.

In the revised manuscript, we modified the schematic in Figure 1b as well as the caption to the figure, to illustrate more clearly which landscape types are addresses in our study. We added the following explanation in the main text:

Note that while our model allows for the formation of surface water bodies through ice-wedge thermokarst, the evolution of already existing old thaw lakes is not considered in this study.

We furthermore added a sentence discussing the importance of thermokarst lakes for carbon mobilization:

We note that rapid thawing of permafrost beneath existing thermokarst lakes was not considered in this study, but constitutes another pathway of unlocking frozen carbon stocks that is typically ignored by ESMS [Langer et al. (2016), Walter Anthony et al. (2018)].

3) Walter Anthony et al. (2018) found that RCP8.5 warming affected yedoma carbon release very differently than RCP4.5 warming because of more rapid thaw lake drainage under RCP8.5. This is a mechanism by which thaw lake carbon mobilization could be greater under mitigation scenarios relative to RCP8.5 warming scenarios. It is not clear to me whether this study is able to address this issue. If not, how much of the results of this study are likely to change if thaw lake drainage is included more explicitly?

As mentioned in the reply to point 1), our model does not explicitly simulate the spatio-temporal dynamics of thermokarst or thermoerosion landforms. Hence, lake drainage events, which result from incision of thaw lake basins by thermo-erosional features, are not explicitly represented in our simulations.

Walter Anthony et al. (2018) accounted for thaw lake dynamics by assuming a warming optimum of 4-6°C for the abundance of thermokarst lakes. This optimum warming level is exceeded in many locations under RCP8.5 warming, but not under RCP4.5 warming, resulting in the above-mentioned effect. It should be noted, however, that this “working hypothesis” of Walter Anthony et al. was partly motivated by the occurrence of vertical subsurface drainage pathways by formation of through taliks in the discontinuous permafrost. In our study area, which is characterized by deep-reaching continuous permafrost, and for which precipitation is projected to exceed evapotranspiration, drainage of lakes occurs mainly laterally, i.e. through incision of lake basins by thermo-erosion features, while vertical lake drainage is unlikely. It is thus questionable whether the hypothesized effect described by Walter Anthony et al., would be as significant in our study region as it is in their pan-Arctic projections. However, we acknowledge that the meso-scale landscape dynamics exemplified by lake expansion and drainage are important processes to be incorporated into our modeling framework in the future. In addition to the changes mentioned above, we added the following to the revised manuscript text:

In order to further constrain our estimates, more dedicated modelling studies are needed that recognize the spatial variability of the present-day landscape hydrology (which has been affected by thermokarst and thermoerosion in the past), and at the same time realistically reflect the meso-scale hydrological and geomorphological responses to future permafrost degradation, such as the expansion and drainage of thermokarst lakes. Constraining the hydrological response of Arctic lowlands to thawing of permafrost appears even more important when considering that lateral hydrological export might constitute also an important direct pathway for soil carbon loss [Plaza et al. (2019)].

4) The authors present simulation results that in general conform to other modeling studies, but do not compare results from the model to any empirical information. I am not an expert in modeling myself so do not have an expert opinion on what would be considered standard. But it seems important to evaluate the model performance more fully than what is presented in the current manuscript.

It is correct that we do not quantitatively compare model results with observational data in the present study. However, the capability of the CryoGrid 3 model to simulate ground temperatures and thaw depths in the study area has been demonstrated by Westermann et al. (2016), and the set-up for ice-wedge terrain which is central to the present study has extensively been validated by Nitzbon et al. (2019). Nitzbon et al. compared CryoGrid 3 simulations for present-day climate with long-term observations of soil temperatures, thaw depths, water tables, soil moisture levels, snow depths and surface energy fluxes. The study concluded that the model realistically reflects permafrost thaw dynamics moderated by small-scale lateral fluxes in spatially heterogeneous polygonal tundra landscapes. As the focus of the current study is on projecting the landscape evolution and permafrost thaw dynamics under a warming climate, the validation has to rely on present-day observations of the landscapes in question as well as on model-model comparisons. In the present study we compared the simulated landscape evolution with local observations of thaw features, such as drowning ice-wedge polygons, high-centred ice-wedge polygons or thermokarst mounds (“baydzerakhks”) by providing areal photos of these features from the study area. In the revised manuscript we addressed the issues of model evaluation more explicitly:

The correspondence between simulated and observed degradation pathways, together with the comprehensive site-level comparison between simulated and observed permafrost characteristics conducted in a preceding work [Nitzbon et al. (2019)], build confidence in that our simulations represent true “end-members” of landscape evolution in the NESAL under projected twenty-first century climate warming.

We furthermore added the following subsection to the Methods description:

Model evaluation and sensitivity analysis.

*The model set-up for ice-wedge terrain was evaluated against field measurements from a study site in the central Lena River delta [Boike et al. (2019)] in a preceding study [Nitzbon et al. (2019)]. The study showed that simulated soil temperatures, thaw depths, soil moisture levels, water tables, snow heights, and surface energy fluxes for different microtopographic features of ice-wedge terrain (polygon centres and rims) agreed well with observations. The correspondence between simulated and observed degradation pathways and landforms provides further evidence for the validity of our modelling approach (see main text, Figure 2, and Supplementary Figure 7).
[...]*

5) The authors need to use more appropriate terminology in the carbon section. The manuscript currently does a rough calculation of how much permafrost carbon (i.e., a C stock assessment) will be exposed to thaw under different scenarios. This really has nothing to do with mobilization and it is misleading for the authors to use terms like that. In my opinion, this entire section needs to be carefully re-thought.

We thank the reviewer for pointing out this misuse of terminology. To clarify our approach we refrained from using the term “mobilization” in the context of our results in the revised manuscript. Instead, we only speak of “thawed” carbon. We thoroughly revised the entire section addressing the carbon stock assessment. In particular, we modified Figure 4 and put a stronger focus on comparing our simulations for ice-rich terrain with corresponding reference runs which do not represent ice-rich permafrost thaw. Considering the points raised by reviewer #2, we furthermore added results for the RCP2.6 scenario and put our estimates in relation to those of other studies. As we modified the entire section, we do not list all changes hereafter and refer to the revised manuscript instead.

6) The authors make a number of simulations in their modeling framework and simulations, such as decisions regarding aerobic: anaerobic proportions. It seems important for this manuscript to provide an assessment of the sensitivity of model performance to these assumptions and to the suite of parameters involved in the simulations.

We agree with the reviewer that the sensitivity of the modelling results to the assumed parameterizations should be further elaborated on. While the sensitivity to certain parameters has been addressed already in the preceding study by Nitzbon et al. (2019), there are further parameters and newly introduced processes which should be tested in this respect. In the revised manuscript and the supplementary information we present and discuss the results of a sensitivity study, in which various parameters have been varied. We conducted the additional simulations for the Holocene deposits with intermediate excess ice content under RCP4.5 and RCP8.5 as well as under water-logged and well-drained hydrological conditions. We identified the following parameters to be especially interesting to be varied:

- the snow density (ρ_{snow}) because of its strong influence on the ground thermal regime
- the field capacity (θ_{fc}) because of its influence on water table levels and thus the fractions of saturated/unsaturated soil
- the natural porosity (φ_{nat}) of ice-rich deposits, because it sets the fraction of excess ice relative to the total ice content
- the lateral sediment transport coefficients (K_{land} , K_{water}) because they affect the speed of stabilizing processes

In the revised manuscript we added the following to the Methods description:

Model evaluation and sensitivity analysis.

[...]

In order to ensure the robustness of our results and to identify critical processes, we conducted a sensitivity analysis in which single model parameters deviated from their default values. The parameters were related to different processes and included the fresh snow density (ρ_{snow}), the sediment transport coefficients (K_{land} , K_{water}), the field capacity (θ_{fc}), and the natural porosity of ice-rich soil layers (φ_{nat}). Results of the sensitivity analysis are provided in Supplementary Figure 6 and discussed in the Supplementary Methods 4.

We furthermore added extensive description, discussion, and figures of the results to the revised supplementary information (Supplementary Methods 4). These additions are too many to be displayed hereafter and we thus refer to the revised version of the SI.

Overall, the sensitivity analysis confirmed, that while the model projections are subject to uncertainties associated with some of the model parameters, the conclusions of the paper are robust against reasonable variations in these parameters.

We further note that we did not make any assumption regarding aerobic/anaerobic conditions, but rather restricted our model diagnostic to saturated/unsaturated fractions of thawed ground. While these notions are related to each other, we are aware that they are not the same thing. The confusion might have originated from the misuse of the aerobic/anaerobic terminology in the “Diagnostics” section of the Methods. In the revised version of the manuscript we adapted this section:

Maximum thaw depth and fractions of saturated and unsaturated conditions.
Maximum thaw depths are the area-weighted mean of the maximum thaw depth of each tile for each year. The proportion of unsaturated conditions corresponds to the fraction of thawed soil cells (above the permafrost) throughout each year

that contain a non-zero fraction of air; the remaining fraction of thawed cells is saturated. Note that this definition takes into account changing thaw depths and changing water table levels throughout the year.

7) As a more minor point, I do not think that the term “conservation” is appropriate for use in the title of this manuscript. To most scientists, conservation implies active protection or management of the land or resource. What the authors are really hinting at in the title is whether response of ice-rich permafrost will be a slow or fast response, relative to change of ice-poor permafrost.

We agree with this concern and changed the title of the revised manuscript as follows:

Fast response of cold ice-rich permafrost in northeast Siberia to a warming climate

Reviewer #2:

We thank the reviewer for the thorough revision and the encouraging suggestions for our manuscript and address the raised issues below.

[...]

1. The effective hillslope sediment transport model was adapted to tundra environment to model polygonal tundra evolution. Why did authors decide to adapt hillslope sediment transport model instead of adapting one of the existing erosion model by the corresponding rescaling of geological years?

Our model describes the dynamic evolution of the surface topography of polygonal tundra by combining two process parameterizations, which we adapted from previously published literature. The first is the excess ice scheme introduced by Lee et al. (2014) to describe the subsidence of ground upon melting of excess ground ice. The second is a nonlinear hillslope diffusion model by Roering et al. (2001) which was adapted for permafrost environments by Plug and West (2009) as well as Kessler et al. (2012). While the excess ice scheme reflects the degradation of ice-wedge polygons, the hillslope model reflects their stabilization due to deposition of laterally transported sediment on small spatial scales.

In the Methods section of the revised version of our manuscript, we explained our approach more clearly:

The geomorphological evolution of ice-wedge terrain was represented in our model by the combination of two process. First, we used the excess ground ice scheme introduced by Lee et al. [Lee et al. (2014)] which is implemented into CryoGrid 3 [Langer et al. (2016), Westermann et al. (2016)] to simulate the subsidence of the soil surface resulting from melting of excess ground ice. Soil layers with ice contents above the “natural” porosity were considered to contain excess ice. Excess water is moved upwards upon thawing of soil layers containing excess ice, while mineral and organic sediment is routed downwards. This scheme reflects the main process causing the degradation of ice-wedge polygons [Nitzbon et al. (2019)]. Second, we used the non-linear hillslope diffusion scheme by Roering et al. [Roering et al. (2001)] which has been applied to periglacial environments [Plug and West (2009), Kessler et al. (2012)]. This scheme describes the lateral erosion of mineral and organic

sediment which is an important process promoting the stabilization of ice-wedge polygons [Kanevskiy et al. (2017)].
For this study we introduced the lateral sediment transport scheme into CryoGrid 3, by [...]

Existing erosion models for periglacial conditions (e.g., works by Andersen et al. (2015) and Egholm et al. (2015)) parameterize processes such as frost-cracking and frost-creep which control periglacial landscape evolution on timescales of thousands to millions of years. To our opinion, adapting such approaches to the ice-rich lowlands considered in our study by rescaling according to the timescales of interest, is not valid. On timescales of tens to hundreds of years, other processes such as slumping (e.g., Kanevskiy et al. (2017)) are more dominant in environments characterized by ice- and organic-rich soils.

2. Currently, model simulates progressive changes leading transition from one state to another with the corresponding permafrost degradation. From field experiments, we know that LCP can also have small ice wedges in the middle (e.g. studies by Jastrow et al), suggesting changes from HCP to LCP and back. However, it is not clear under what conditions it is possible to cyclically develop a dynamical evolution of HCP to LCP and vice versa?

We agree with the reviewer that the development of polygonal tundra is a cyclic process, and that HCP can evolve back into LCP due to accumulation of ground ice. Kanevskiy et al. (2017) summarized the long-term evolution of ice-wedge polygons very convincingly in a schematic model. However, they suggests that the temporal scales of ice-wedge degradation (sub-decadal or decadal) are typically one or two orders of magnitude smaller than the corresponding temporal scales of ice-wedge stabilization (decades to millenia). Moreover, ice-wedge recovery presumes favorable climatic conditions for ground ice accumulation, i.e. persistent periods of very cold winters.

As our study focuses on ice-wedge development during the 21st century, we assumed that it is valid to neglect ice-wedge growth in our model for two reasons. First, the expected warming for the study region makes the preconditions for sustained ice-wedge growth during the study period unlikely. Second, the temporal scale of ice-wedge recovery is beyond the temporal scales considered in the study.

An improved understanding of past and future permafrost landscape dynamics, however, would require model frameworks that take into account accumulation of ground ice. This is, however, beyond the scope of the present study and the CryoGrid model is not yet capable of simulating accumulation of ground ice.

In the revised version of the manuscript, we discuss this shortcoming of our current framework:

We note that the processes represented in our model are tailored for its application on decadal-to-centennial time-scales. If the approach was applied to millennial time-scales, comprising also extended periods of colder climatic conditions, accumulation of ground ice would need to be taken into account, as the process counter-acts the melting of ice wedges [Kanevskiy et al. (2017)].

Applying the RCP 2.6. scenario might be useful to illustrate how negative trend in the global temperatures during this century could affect changes in polygonal tundra landscapes. It also could be interesting to show that these transitions are possible. For example, Kleinen and Brovkin (2018) drew some insightful conclusions based RCP 2.6 suggesting that under this scenario almost double amount of carbon need to be removed from the atmosphere in order to reach the 1.5°C target.

We agree that considering the RCP2.6 scenario is an interesting addition to our study, particularly in the context of the 1.5° target. We thus conducted additional simulations under RCP2.6 to provide a more complete picture of possible pathways of permafrost degradation and thaw-affected carbon stocks in the study region.

In the revised versions of the manuscript and the supplementary material we added descriptions of the methodology, results, discussion, and figures related to the RCP2.6 simulations. We especially considered the RCP2.6 simulations in the context of organic carbon stocks subject to thaw and included them in the revised version of Figure 4. The changes to the manuscript with respect to the RCP2.6 simulations include, but are not limit to the following:

Results and Discussion

[...]

Under the RCP2.6 scenario all landscape types (LB, HD, YD) remained stable throughout the simulation period, with the exception water-logged YD where shallow surface water bodies formed. We thus restrict the following analysis of the landscape evolution to the warming scenarios RCP4.5 and RCP8.5.

[...]

Simulations for the ambitious mitigation scenario RCP2.6 revealed that ice-rich permafrost remained largely stable throughout the simulation period (Supplementary Figure 10).

[...]

Under the ambitious mitigation scenario RCP2.6, no widespread occurrence of excess ice melt was simulated (Supplementary Figure 10), and consequently the amounts of carbon affected by thaw by 2100 (0.8-2.0 GtC) did not exceed the respective projection of the reference run (1.3 GtC).

[...]

Methods

[...]

For the RCP2.6 scenario, we took anomalies from CCSM4 projections and applied them repeatedly to the period from 2000 until 2014 of the RCP4.5 forcing data.

Further changes can be seen in the “change-highlighted” versions of the revised manuscript and supplementary information.

With our model set-up, it was, however, not possible to simulate the transition from HCP to LCP under such a scenario, as the necessary processes such as frost-cracking and ground-heave are not implemented in the CryoGrid 3 model, and the considered time-scales of our study are most likely too short to capture a full cycle of polygon development (see previous response to the first part of point 2).

3. Finally, authors used simplified empirical relationship to address the amount carbon that could be released as a result of permafrost thawing. Recent studies show that observation from flux towers indicate that most of this release is not captured by the near-by flux towers, suggesting significant lateral transport of the carbon in the form of the dissolved organic carbon (Plaza et al. 2019). Taking this into account how it might change the current numbers on Figure 4.

We would like to point out, that our study quantifies the amounts of carbon which would become subject to thawed conditions, while it does not assess fluxes of carbon into either the atmosphere or via lateral export. The numbers presented in Figure 4 reflect the amounts of carbon that would potentially become available for any mobilization pathway, either directly into the atmosphere, or through lateral erosion. The model we used, however, does not allow

for a quantification of actual carbon mobilization pathways.

In the revised version of the manuscript, we clarified which processes are (not) represented by our model, and we carefully revised the terminology with respect to carbon mobilization.

As the studies of Kleinen and Brovkin (2018) and Plaza et al. (2019) provide relevant results in the context of our assessment of vulnerable carbon, we discussed their results in the revised manuscript.

Authors claim that there is such a strong dependence between subsidence, amount of ground ice, and carbon emissions. I suggest to include plots illustrating changes in the carbon emission with respect to subsidence and ground ice. This might to conceptualize these relationships and benefit future studies working on similar topic.

We appreciate this suggestion very much and elaborated on the relation between ground ice, permafrost thaw, subsidence and vulnerable carbon in the revised version of the supplementary information. There we added an entire section as well as two plots (Supplementary Figure 13) illustrating these relations. We further provide an equation which relates permafrost thaw and ground subsidence in dependence of excess ice content under the idealized assumptions of our simulations.

In the main text of the revised manuscript, we mention these relations and discuss the idealized assumptions:

While a simple linear relation between total permafrost degradation and excess ground ice content can be established under the idealized assumptions of our model (see Supplementary Methods 6 and Supplementary Figure 13), the actual timing and the temporal evolution of permafrost thaw is further affected by site-specific factors, the incidence of extreme weather conditions, as well as the hydrophysical and ecological feedback processes.

From page 8. “Locally observed degradation” It is not clear does the model address the effect of the anthropogenic origins or not?

Our model does not address direct perturbations of the permafrost of anthropogenic origin. However, such local perturbations might initiate ice-wedge degradation already today at locations where the permafrost otherwise would still be stable. Hence, we argue that the reproduction of such “locally observed degradation” features with our model builds confidence in our approach and parameterization.

In the revised version of the manuscript, we revised the respective session addressing disturbances of anthropogenic origin:

Within the NESAL, such thaw phenomena do occur locally [Liljedahl et al. (2016)], but are either restricted to extreme site-specific conditions or induced by disturbances of natural or anthropogenic origin. Examples include the accumulation of snow in topographic depressions [Liljedahl et al. (2016), Abolt et al. (2018)], the removal of protective organic layers or vegetation [Nauta et al. (2015)], tundra and forest fires [Jones et al. (2015)], or vehicle tracks. When introducing such conditions or disturbances into our model, the simulated timing of permafrost degradation markedly shifts to earlier years [Nitzbon et al. (2019)] – in agreement with observations [Liljedahl et al. (2016)]. Here, however, we concentrated on undisturbed initial conditions and evaluate the impact of climate warming.

It would be nice to know the percent difference between projected carbon release by global models from Russian permafrost and current study only for tundra over 21st century. Showing this difference can make a greater impact and better convince readers about importance of the tundra (yedoma) permafrost thaw.

We appreciate this suggestion and agree that such a comparison would emphasize the importance of ice-rich permafrost thaw in the context of the global carbon cycle. While we could not identify suitable publications assessing vulnerable carbon stocks from Russian permafrost, we included comparisons to pan-Arctic assessments of vulnerable carbon by models which lack representation of deep carbon stocks and of thaw processes acting in ice-rich permafrost.

Generally, in the revised version we put a stronger emphasis on the comparison between our model projections for ice-rich ground and the reference runs with a more simplistic representation of permafrost (e.g., Figure 4). This way, the relevance of thaw processes in ice-rich permafrost is communicated more clearly. Changes to the manuscript:

These results suggest that in regions like the NESAL which host cold, ice- and organic-rich permafrost deposits, substantially larger amounts of carbon could thaw than they are projected by models like ESMs which employ a simplistic representation of permafrost thaw dynamics. Such global-scale models project significant amounts of thawed permafrost carbon for the end of the twenty-first century (about 140–400 GtC, depending on the scenario and the model [Burke et al. (2012), Kleinen and Brovkin (2018)]), but do not take into account rapid thaw processes in ice-rich permafrost, and also ignore deep carbon stocks becoming accessible through these processes. This is particularly problematic for cold permafrost regions like the NESAL where – even under RCP8.5 – projected gradual thaw is limited (only about 5.3% of the NESAL's carbon pools become subject to thaw in our reference runs), but rapid thaw of ice-rich permafrost could unlock 2–12 times (i.e., up to two-thirds of the NESAL's carbon pool) of the amount projected for gradual thaw only. ESMs might thus substantially underestimate the amounts of carbon becoming accessible for microbial decomposition under a warming climate, particularly in cold and seemingly stable permafrost regions like the NESAL.

Additional References

Andersen, J. L., Egholm, D. L., Knudsen, M. F., Jansen, J. D., & Nielsen, S. B. (2015). The periglacial engine of mountain erosion – Part 1: Rates of frost cracking and frost creep. *Earth Surface Dynamics*, 3(4), 447–462. <https://doi.org/10.5194/esurf-3-447-2015>

Egholm, D. L., Andersen, J. L., Knudsen, M. F., Jansen, J. D., & Nielsen, S. B. (2015). The periglacial engine of mountain erosion – Part 2: Modelling large-scale landscape evolution. *Earth Surface Dynamics*, 3(4), 463–482. <https://doi.org/10.5194/esurf-3-463-2015>

Fritz, M., Opel, T., Tanski, G., Herzsuh, U., Meyer, H., Eulenburg, A., & Lantuit, H. (2015). Dissolved organic carbon (DOC) in Arctic ground ice. *The Cryosphere*, 9(2), 737–752. <https://doi.org/10.5194/tc-9-737-2015>

REVIEWERS' COMMENTS:

Reviewer #1 (Remarks to the Author):

This is the second time I have reviewed this manuscript and I found the revised manuscript to be much improved as a result of the revision process. The authors should be commended for taking reviewer comments seriously and addressing issues both through text revisions but also in some cases by including new supplemental information.

After evaluating the individual responses to reviewer comments, I do note that the authors spent considerable effort responding to each comment, but some of this information remained in the responses rather than being incorporated into the manuscript text. I do think the revised manuscript overall is acceptable for publication, but there are some interesting points that did not actually get incorporated into the manuscript. I would encourage the authors to reread the response document and consider adding some additional information from these responses into the manuscript or the supplemental file. A reader is very well likely to benefit from this information.

A few minor points regarding the treatment of carbon stocks:

1) at the bottom of page 16, "substantially larger amounts of carbon could thaw". While I understand the key point here, I wanted to point out that it's not the carbon per se that is thawing. The authors could clarify that the carbon rich soils are thawing, or that permafrost organic matter is thawing, exposing carbon to mineralization.

2) Somewhere perhaps on page 18 or page 19, I think the authors need to point out that carbon emissions to the atmosphere are not determined solely by thaw and mineralization of soil/sediment carbon. Rather measurements of ecosystem carbon exchange also need to include assessments of carbon uptake by vegetation. Not only would this more accurately reflect the measurements/assessments that are required for models to address carbon balance/exchange from changing permafrost environments, but it will help to reinforce/remind readers that they cannot assume that the carbon stock estimates presented here will all wind up in the atmosphere. A well placed sentence or two could solidify this concept.

3) Finally, there is a running debate on social media between climate and energy experts regarding RCP8.5. The argument has gone back and forth for months, but boils down to a segment of the community arguing that we cannot and should not refer to RCP8.5 as "business-as-usual", which the authors do at the end of the paper (and perhaps in other places?). The debate is too lengthy and nuanced to summarize here. In the end, the authors have done a nice job in this study comparing different RCP scenarios here, and this is exactly what is needed. However, I would encourage the authors to strictly refer to the scenarios by name, and avoid terms like "business-as-usual".

Reviewer #2 (Remarks to the Author):

Major

The current version of the manuscript was significantly improved. The overall mechanics of the ice-wedge dynamics require some further clarification. In particular, it was not clear to me, how does the thermodynamics between air and surface water work? Once the ice-wedge is melted and water starts ponding at the surface. If the water body completely freezes over the winter and snow depth is shallow then it could lead to colder subsurface contributing to the retardation of the permafrost thawing. Also, when the standing water is accumulated, how exactly does CryoGrid model treat the standing water physics? Does the water layer has water properties or properties of subsurface modified to represent fully saturated soil with thermal properties corresponding to water?

I apologize if I missed that but the paper is refereeing to rapid thaw process/features without explicitly defining it. To me instead of calling it rapid thaw, which I found confusing, it would be better to say that these processes are micro-scale processes and build your story of that, instead of referring to "rapid thaw". The micro-scale processes make a perfect sense to readers and it is clear that they are not represented in any LSM or GCM type models. The introduction of the micro-scale process (ice-wedge dynamics) allows us to better address subsurface carbon dynamics in the tundra. In addition, it would be useful talking about when the tundra would switch from carbon sink to carbon source. Using the language that is more in-line with the PCN language (e.g. McGuire et al., 2018). That said, I would encourage authors not to refer to the "rapid thaw" process. Instead, the scale is the most important feature here that is not represented in global-scale models.

Minor

Why the authors decided to average over 11 years? Why not 5 or 10? Need to explain that

SI. P10. L6 change 1999 to 2099

Main. P3.L2. change "very" to "high".

Consider changing Fig 7b with the one that is not from the coast because the current one illustrates coastal erosion and could be confusing. Since the story has nothing to do with coastal erosion.

How exactly lateral heat and fluid flow happening between adjacent polygons (SI Fig 3a)?

Also, I would suggest focusing on the SI and add more details to the description of the thermo-hydro-mechanical processes.

Please find below responses to the issues raised by the two reviewers. Reviewer comments are in **boldface** and extracts from the manuscript are in *italics*, changes to the manuscript are highlighted **yellow**.

All references used in the responses below are either contained in the reference list of the revised manuscript or provided at the end of this response.

We also provide a change-highlighted version of the revised manuscript where all changes compared to the previous submission are visualized.

Reviewer #1:

[...]

After evaluating the individual responses to reviewer comments, I do note that the authors spent considerable effort responding to each comment, but some of this information remained in the responses rather than being incorporated into the manuscript text. I do think the revised manuscript overall is acceptable for publication, but there are some interesting points that did not actually get incorporated into the manuscript. I would encourage the authors to reread the response document and consider adding some additional information from these responses into the manuscript or the supplemental file. A reader is very well likely to benefit from this information.

We thank the reviewer for this suggestion. We incorporated further information from the responses into the main text and the supplementary information. For example, since many reviewer comments could be addressed by clarifying the scope and justifying the approach of our study, we complemented the Supplementary Information by two additional sections addressing the “Terminology for permafrost thaw processes” (Supplementary Notes 1) and the “Dominant types of excess ground ice and associated degradation pathways” (Supplementary Notes 2). Further changes to the main text and the supplementary information can be seen in the “change-highlighted” versions provided with the final revision.

1) at the bottom of page 16, "substantially larger amounts of carbon could thaw". While I understand the key point here, I wanted to point out that it's not the carbon per se that is thawing. The authors could clarify that the carbon rich soils are thawing, or that permafrost organic matter is thawing, exposing carbon to mineralization.

We agree with this imprecise use of terminology and changed the text accordingly:

*[...], substantially larger amounts of **permafrost organic matter could thaw and expose carbon to mineralization** than what is projected by models like ESMs which employ a simplistic representation of permafrost thaw dynamics. Such global-scale models project significant amounts of **thaw-affected permafrost carbon** for the end of the twenty-first century [...]*

2) Somewhere perhaps on page 18 or page 19, I think the authors need to point out that carbon emissions to the atmosphere are not determined solely by thaw and mineralization of soil/sediment carbon. Rather measurements of ecosystem carbon exchange also need to include assessments of carbon uptake by vegetation. Not only would this more accurately reflect the measurements/assessments that are required for models to address carbon balance/exchange from changing permafrost environments, but it will help to reinforce/remind readers that they cannot assume that the carbon stock estimates presented here will all wind up in the atmosphere. A well placed sentence or two could solidify this concept.

We thank the reviewer for this important suggestion for precluding misinterpretations of our findings. We added the following sentences in order to clarify our approach:

It should be stressed, that the overall carbon balance of permafrost ecosystems under a changing climate is depending on various processes besides permafrost thaw, including, for example, carbon uptake by vegetation. Our simulations do not allow conclusions about which portion of the thaw-affected carbon stocks presented in Figure 4 would end up as greenhouse gases in the atmosphere, and whether the study region would evolve into a carbon source or sink.

3) Finally, there is a running debate on social media between climate and energy experts regarding RCP8.5. The argument has gone back and forth for months, but boils down to a segment of the community arguing that we cannot and should not refer to RCP8.5 as "business-as-usual", which the authors do at the end of the paper (and perhaps in other places?). The debate is too lengthy and nuanced to summarize here. In the end, the authors have done a nice job in this study comparing different RCP scenarios here, and this is exactly what is needed. However, I would encourage the authors to strictly refer to the scenarios by name, and avoid terms like "business-as-usual".

We thank the reviewer for pointing this out and agree with the concerns of using the term "business-as-usual". We changed the manuscript accordingly and refer to the scenario by name:

[...] under a high emissions scenario (RCP8.5).

Reviewer #2:

The current version of the manuscript was significantly improved. The overall mechanics of the ice-wedge dynamics require some further clarification. In particular, it was not clear to me, how does the thermodynamics between air and surface water work? Once the ice-wedge is melted and water starts ponding at the surface. If the water body completely freezes over the winter and snow depth is shallow then it could lead to colder subsurface contributing to the retardation of the permafrost thawing.

We thank the reviewer for pointing out shortcomings of our model description.

The surface energy balance scheme of CryoGrid distinguished between different types of surfaces (e.g. soil, surface water, snow) to allow realistic calculations of water and energy exchange between the atmosphere and the ground. CryoGrid does allow surface water bodies to freeze completely, allowing for efficient export of heat from the ground to the atmosphere. The maximum snow depth was limited to 0.4m above the tile with the highest relative elevation. While this guarantees a limitation of the maximum snow depth atop frozen water surfaces, we did not take into account the potential reduction of snow cover through wind blow atop of large water bodies. At the same time, we note, that for this wind-blow effect, sufficiently large frozen water bodies are needed, which are likely to be too deep to freeze through in winter, particularly under climate warming scenarios.

In the final revised version we mention the effect of blowing snow from frozen lake surfaces in the context of various other meso-scale landscape feedbacks:

[...], more dedicated modelling studies are needed that recognize the spatial variability of the present-day landscapes' topography and hydrology [...] Meso-scale snow redistribution (e.g. blowing snow from frozen lake surfaces)

constitutes another process which can affect the thermal and hydrological state and thaw dynamics of permafrost.

In addition, we clarified the snow redistribution scheme and mention the limitation of the maximum snow depth in the final revised version of the SI:

Snow was redistributed among the different tiles (polygon centres, polygon rims, and troughs), assuming a preferential accumulation of snow in topographic depressions. The maximum height of the snow pack was limited to $h^{max}=0.4$ m relative to the tile with the highest topographic elevation (including soil and water surfaces).

Also, when the standing water is accumulated, how exactly does CryoGrid model treat the standing water physics? Does the water layer has water properties or properties of subsurface modified to represent fully saturated soil with thermal properties corresponding to water?

CryoGrid 3 assumes instantaneous mixing of surface water during ice-free conditions to simulate the heat transport within standing surface water, i.e. heat transfer occurs convective rather than conductive. This approach is supported by field measurements (Boike et al. (2015)) and has been described and evaluated in preceding studies using CryoGrid 3 (Westermann et al. (2016), Nitzbon et al. (2019)).

I apologize if I missed that but the paper is refereeing to rapid thaw process/features without explicitly defining it. To me instead of calling it rapid thaw, which I found confusing, it would be better to say that these processes are micro-scale processes and build your story of that, instead of referring to “rapid thaw”. The micro-scale processes make a perfect sense to readers and it is clear that they are not represented in any LSM or GCM type models. The introduction of the micro-scale process (ice-wedge dynamics) allows us to better address subsurface carbon dynamics in the tundra. In addition, it would be useful talking about when the tundra would switch from carbon sink to carbon source. Using the language that is more in-line with the PCN language (e.g. McGuire et al., 2018). That said, I would encourage authors not to refer to the “rapid thaw” process. Instead, the scale is the most important feature here that is not represented in global-scale models.

We thank the reviewer for pointing out the lack of a definition for our terminology and for suggesting an alternative framing. For us, the term “rapid thaw” is synonym to “abrupt thaw” which has been introduced to characterize thawing of ice-rich permafrost, which is fundamentally different from gradual (top-down) thawing of permafrost (Schuur et al. (2015), Turetsky et al. (2019), Turetsky et al. (2020)). It is important to note, that it is not only the micro-scale character of the thaw process, but that the thawing of (heterogeneously distributed) excess ground ice induces landscape change and feedbacks, which are collectively denoted as thermokarst. We acknowledge, that the terms “rapid” and “abrupt” suggest certain temporal characteristics of the thaw process which are subjective and thus do not necessarily correspond to the notions of field scientists. For the final revision of our manuscript, we thus carefully reconsidered usage of the term “rapid thaw” in our study. Given the above consideration we decided to use the term “thermokarst-inducing processes” to refer to structures and processes which are needed for representing the initiation and evolution of thermokarst and associated feedbacks in numerical permafrost models. This term is more descriptive than “rapid thaw” as it does not imply any notion of the temporal characteristic of the process. Furthermore, it connects to the established notion of “thermokarst” used by field scientist, while being useful to characterize numerical permafrost models.

In the final revised manuscript, modified some formulations in the Abstract and Introduction, in order to introduce a consistent terminology for our study.

Thawing of ice-rich permafrost and melting of massive ground ice induce landscape change termed thermokarst, which results in characteristic landforms across ice-rich permafrost terrain (Kokelj 2013) (see Supplementary Notes 1 for definitions). In the continuous permafrost zone, thermokarst is expressed in the transition from low-centred to high-centred ice-wedge polygons (Jorgenson et al. 2006, Liljedahl et al. 2016), or the formation of thaw lakes and thermo-erosional gullies (Kokelj et al. (2013), Olefeldt et al. (2016), Walter Anthony et al. (2018)), thereby causing landscape-scale feedbacks on hydrology and carbon decomposition (Lara et al. (2015), Liljedahl et al. (2016), Walvoord et al. (2016)). In contrast to the gradual thawing of permafrost in ice-poor terrain, thermokarst processes can cause severe permafrost degradation within few years or decades, and have thus been referred to as rapid or abrupt thaw (Schuur et al. (2015), Turetsky et al. (2019), Turetsky et al. (2020)). The contribution of thermokarst processes to global-scale permafrost degradation in the future is highly uncertain (Schuur et al. (2015), Turetsky et al. (2019)).

Furthermore, we adapted formulations containing the term “rapid thaw” so that they align with the revised terminology (see “change-highlighted” version).

We also complemented the Supplementary Information by a section addressing the terminology and giving definitions of potentially ambiguous terms.

With respect to the timing of a transition of the tundra from a carbon source to a sink, we would like to point out that our simulations do not allow conclusions about the full carbon balance of the investigated ecosystems. This limitation is stressed in the final revised article (see the reply to point 2 of reviewer #1).

Why the authors decided to average over 11 years? Why not 5 or 10? Need to explain that

We chose an odd number such that the displayed average is symmetric around the respective year. We found an 11-year period (instead of, e.g., a 5-year period) to be a good compromise to visualize long-term trends without overly emphasizing sub-decadal variability.

SI. P10. L6 change 1999 to 2099

Thanks. Changed.

Main. P3.L2. change “very” to “high”.

We dropped the word as it was not important at this point.

Consider changing Fig 7b with the one that is not from the coast because the current one illustrates coastal erosion and could be confusing. Since the story has nothing to do with coastal erosion.

We complemented Fig. 7 by an aerial image of thermokarst mounds (“baidzharaks”) (new Fig. 7d) which illustrate these landforms as a spatial feature rather than at an exposed shoreline.

We decided to keep Fig. 7b as it gives an impression of the inactive surface of Yedoma deposits in the study region.

How exactly lateral heat and fluid flow happening between adjacent polygons (SI Fig 3a)?

For our simulations we assumed that ice-wedges polygons are symmetric and arranged in periodic way, i.e., the simulations of one ice-wedge polygon effectively represent an entire part of the landscape (Nitzbon et al., 2019). Hence, there is not explicit representation of lateral heat and water fluxes between adjacent polygons in our model.

Also, I would suggest focusing on the SI and add more details to the description of the thermo-hydro-mechanical processes.

For the final revised version of the article, we restructured the supplementary information and added further background information. As we based our study largely on model version which have been described and evaluated in preceding publications (e.g., Westermann et al. (2016), Langer et al. (2016), Nitzbon et al. (2019)), we focused on the description of new model components and changes compared to previous model versions. For details on the process parameterizations we refer to the studies in which they have been introduced into CryoGrid.

Additional references

Boike, J., Georgi, C., Kirilin, G., Muster, S., Abramova, K., Fedorova, I., Chetverova, A., Grigoriev, M., Bornemann, N., & Langer, M. (2015). Thermal processes of thermokarst lakes in the continuous permafrost zone of northern Siberia – observations and modeling (Lena River Delta, Siberia). *Biogeosciences*, 12(20), 5941–5965.
<https://doi.org/10.5194/bg-12-5941-2015>